# A Cross-Domain Benchmark for Active Learning

**Thorben Werner** *
University of Hildesheim
Universitätsplatz 1  31141 Hildesheim
`werner@ismll.de`

**Johannes Burchert**\*
University of Hildesheim
Universitätsplatz 1, 31141 Hildesheim
`burchert@ismll.de`

**Maximilian Stubbemann**\*
University of Hildesheim
Universitätsplatz 1, 31141 Hildesheim
`stubbemann@ismll.de`

**Lars Schmidt-Thieme**\*
University of Hildesheim
Universitätsplatz 1, 31141 Hildesheim
`schmidt-thieme@ismll.uni-hildesheim.de`

## Abstract

Active Learning (AL) deals with identifying the most informative samples for labeling to reduce data annotation costs for supervised learning tasks. AL research suffers from the fact that lifts from literature generalize poorly and that only a small number of repetitions of experiments are conducted. To overcome these obstacles, we propose *CDALBench*, the first active learning benchmark which includes tasks in computer vision, natural language processing and tabular learning. Furthermore, by providing an efficient, greedy oracle, *CDALBench* can be evaluated with 50 runs for each experiment. We show, that both the cross-domain character and a large amount of repetitions are crucial for sophisticated evaluation of AL research. Concretely, we show that the superiority of specific methods varies over the different domains, making it important to evaluate Active Learning with a cross-domain benchmark. Additionally, we show that having a large amount of runs is crucial. With only conducting three runs as often done in the literature, the superiority of specific methods can strongly vary with the specific runs. This effect is so strong, that, depending on the seed, even a well-established method's performance can be significantly better and significantly worse than random for the same dataset.

## 1 Introduction

Deep neural networks (NN) have produced state-of-the-art results on many important supervised learning tasks. Since Deep NNs usually require large amounts of labeled training data, Active Learning (AL) deals with selecting the most informative samples out of a large pool of unlabeled data, so that only these samples need to be labeled. It has been shown that a small labeled set of this nature can be used to train well-performing models. In the last decade, many different algorithms for AL have been proposed and almost every method has reported lifts over every single one of its

---

*Institute of Computer Science - Information Systems and Machine Learning Lab (ISMLL)

38th Conference on Neural Information Processing Systems (NeurIPS 2024) Track on Datasets and Benchmarks.

Table 1: Comparison of our benchmark with the existing literature. Oracle curves serve as an approximation of the best possible AL algorithm. Our benchmark contains 9 datasets (14 including the encoded versions). "Semi" indicates whether the paper is employing any form of self- or semi-supervised learning. A "-" for repetitions means that we could not determine how often each experiment was repeated in the respective framework. *CDALBench* is the only benchmark which evaluates a high number of runs and considers all 5 domains.

| Paper | Sampling | #Data | #Alg | Img | Txt | Tab | Synth | Semi | Oracle | Repetitions |
|---|---|---|---|---|---|---|---|---|---|---|
| Beck et al. [3] | batch | 4 | 7 | ✓ | - | - | - | - | - | - |
| Hu et al. [11] | batch | 5 | 13 | ✓ | ✓ | - | - | - | - | 3 |
| Zhou et al. [34] | batch | 3 | 2 | ✓ | ✓ | - | - | - | ✓ | 5 |
| Zhan et al. [30] | single+batch | 35 | 18 | - | - | ✓ | ✓ | - | ✓ | 10-100 |
| Munjal et al. [22] | batch | 2 | 8 | ✓ | - | - | - | - | - | 3 |
| Li et al. [18] | batch | 5 | 13 | ✓ | - | - | - | ✓ | - | - |
| Rauch et al. [25] | batch | 11 | 5 | - | ✓ | - | - | - | - | 5 |
| Zhang et al. [32] | batch | 6 | 7 | ✓ | - | - | - | - | - | 2-4 |
| Bahri et al. [2] | batch | 69 | 16 | - | - | ✓ | - | - | - | 2-4 |
| Ji et al. [13] | batch | 3 | 8 | ✓ | - | - | - | - | - | - |
| Lueth et al. [20] | batch | 4 | 5 | ✓ | - | - | - | ✓ | - | 3 |
| **Ours** | single+batch | 9(14) | 11 | ✓ | ✓ | ✓ | ✓ | ✓ | ✓ | 50 |

predecessors. [2] However, real insights into the current state of AL are hard to draw from these works, due to the following reasons: 1. These works do not use a standardized evaluation setting with fixed datasets and baseline approaches. 2. Due to computational constraints, a lot of works perform only a small amount of experimental runs, hence it is questionable wether the superiority of a specific approach can be concluded from the conducted experiments. 3. The works are only evaluated in a specific domain, such as computer vision or language processing. However, AL is a general principle of supervised learning, and thus methods should be evaluated in multiple domains to assess their capabilities.

While multiple benchmark suites have been proposed to solve problem 1, to the best of our knowledge, all of them are either limited in the domains they consider or do not contain enough runs to generate conclusive results. Hence, the current SOTA in AL is still not well-understood and principled shortcomings of different algorithms and wether they are domain-independent are currently not identified.

Here we step in with *CDALBench*, an AL benchmark which covers multiple application domains and reports a large amount of runs per experiment, so that the significance of performance differences can be estimated. Specifically, *CDALBench* consists of datasets from computer vision, natural language processing and the tabular domain. We provide our datasets both in normal format as well as "embedded" by a fixed embedding model, enabling evaluation of AL methods in this semi-supervised setting. Furthermore, we propose two novel synthetic datasets to highlight general challenges for AL methods. The applied evaluation protocol in *CDALBench* uses 50 runs for each experiment. By having such a large amount of runs, we can evaluate the significance of performance gaps and identify the best performing approaches for each dataset as well as whole domains. Furthermore, we show that the small amount of runs other works do, in fact, produce misleading results. To be more specific, we show that if only 3 restarts are employed for each experiment, the performance of specific methods strongly varies. As we will see, even the ranking of the different methods averaged over many datasets fluctuates with the specific set of runs. This effect is so strong, that, depending on the seed, even a well-established method's performance can be significantly better and significantly worse than random for the same dataset.

To enable the computation of an oracle performance for a protocol with large amounts of restarts, we propose a *greedy oracle algorithm* which uses only a small amount of search steps to estimate the optimal solution. Our oracle relies on directly testing a small sample of points in every iteration whether they induce an improvement in test accuracy and selects the optimal point from that small sample. While being more time-efficient than established oracle functions, it possibly underestimates the real upper bound performance. However, as our experiments will show, it is still outperforming all current AL methods by at least 5% and thus is suitable as an upper bound.

---

[2]Out of all considered algorithms for this paper, only BALD [9] did not claim a new state-of-the-art (SOTA) performance in their result section.

Our experimental evaluation shows that there exists no clear SOTA method for AL. The superiority of methods is strongly dataset- and domain-dependent with the outstanding observation, that the image domain works fundamentally different than the tabular and text domain. Here, the best performing approach for text and tabular data, namely *margin sampling*, is significantly outperformed by *least confident sampling*, which does not belong to the top performing approaches in any other domain. Thus, using the performance of AL approaches on the image domain as a proxy of AL in general, as it is often done [3, 22, 18, 13, 20], is questionable. To further analyze performance of common methods, we propose *Honeypot* and *Diverging Sine*, two synthetic datasets, designed to be challenging for naive decision-boundary- and clustering-based approaches respectively. Hence, they provide insights in principled shortcomings of AL methods.

In summary, *CDALBench* is an experimental framework which includes an efficient oracle approximation, multiple application domains, enough repetitions to draw valid conclusions and two synthetic tasks to highlight shortcomings of AL methods. By being the first benchmark to providing these points in one code-base, we believe that *CDALBench* is a major step forward of assessing the overall state of AL research, independent of specific application domains. *CDALBench* is publicly available under `https://github.com/wernerth94/A-Cross-Domain-Benchmark-for-Active-Learning/`.

Our contributions include the following:

1. We show that the small number of repetitions that previous works have employed is not sufficient for meaningful conclusions. Sometimes even making it impossible to assess if a performance is above or below random.

2. We propose an efficient and performant oracle which is constructed iteratively in a greedy fashion, overcoming major computational hurdles.

3. We propose *CDALBench*, the first general benchmark providing tasks in the domains of image, text and tabular learning. It further contains synthetic and pre-encoded data to allow for a sophisticated evaluation of AL methods. Our experiments show, that there is no clear SOTA method for AL across different domains.

4. We propose *Honeypot* and *Diverging Sin*, two synthetic datasets designed to hinder AL by naive decision-boundary- or clustering-based approaches respectively. Thus, they provide an important tool to identify shortcomings of existing AL methods.

## 2 Problem Description

Given two spaces $\mathcal{X}, \mathcal{Y}$, $n = l + u$ data points with $l \in \mathbb{N}$ labeled examples $\mathcal{L} = \{(x_1, y_1), \ldots, (x_l, y_l)\}$, $u \in \mathbb{N}$ unlabeled examples $\mathcal{U} = \{x_{l+1}, \ldots, x_n\}$, a model $\hat{y} : \mathcal{X} \to \mathcal{Y}$, a budget $\mathbb{N} \ni b \leq u$ and an annotator $A : \mathcal{X} \to \mathcal{Y}$ that can label $x$. We call $x \in \mathcal{X}, y \in \mathcal{Y}$ predictors and labels respectively where $(x, y)$ are drawn from an unknown distribution $\rho$. Find an AL method $\Omega : \mathcal{U}^{(i)}, \mathcal{L}^{(i)} \mapsto x^{(i)} \in \mathcal{U}^{(i)}$ that iteratively selects the next unlabeled point $x^{(i)}$ for labeling

$$\mathcal{L}^{(i+1)} \leftarrow \mathcal{L}^{(i)} \cup \left\{ \left( x^{(i)}, A(x^{(i)}) \right) \right\}$$

$$\mathcal{U}^{(i+1)} \leftarrow \mathcal{U}^{(i)} \setminus \left\{ x^{(i)} \right\}$$

with $\mathcal{U}^{(0)} = \text{seed}(\mathcal{U}, s)$ and $\mathcal{L}^{(0)} = \left( \mathcal{U}_i^{(0)}, A(\mathcal{U}_i^{(0)}) \right)$ $i \in [1, \ldots, s]$, where $\text{seed}(\mathcal{U}, s)$ selects $s$ points per class for the initial labeled set $\mathcal{L}^{(0)}$.

So that the average expected loss $\ell : \mathcal{Y} \times \mathcal{Y} \to \mathbb{R}$ of a machine learning algorithm fitting $\hat{y}^{(i)}$ on the respective labeled set $\mathcal{L}^{(i)}$ is minimal:

$$\min \quad \frac{1}{B} \sum_{i=0}^{B} \mathbb{E}_{(x,y) \sim \rho} \ell(y, \hat{y}^{(i)})$$

## 3 Related Work

While multiple benchmark suites have been proposed for AL, none of them provide experiments for more than two domains. The authors of [3], [22], [18], [13] and [20] even focus exclusively

Figure 1: Random draws from a pool of 100 runs for margin sampling on the Splice dataset with different numbers of repetitions ($\alpha = \{3, 5, 50\}$). Green curves are the mean performance of all 100 runs, while the samples are blue. Even with 3 or 5 repetitions, we can observe that single draws for margin sampling display below-random performance (black), while the true mean should be above random.

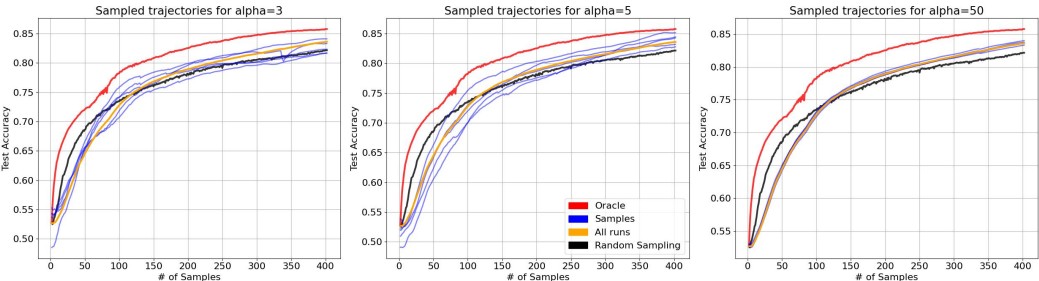

on the image domain. Especially the tabular domain is underrepresented in preceding benchmarks, as only [31] provides experiments for it. The interplay between AL and semi-supervised learning is similarly under-researched, as only two works exist [18, 20], both of them only using images. An oracle algorithm has been proposed by two works [34, 31]. Both of these algorithms rely on search and are computationally very expensive, while our proposed method efficiently can be constructed sequentially. The two closest related works to this benchmark are [13] and [20], who also place a much higher emphasis on the problem of evaluating AL methods under high variance than their predecessors (indicated in Tab. 1 by a dashed line). The authors of [13] posed a total of 11 "recommendations" for reliable evaluation of AL methods. We largely adapt the proposed recommendations and extend their work to multiple domains and query sizes. For a complete list of the recommendations and our implementation of them, please refer to App. A. This work also pays attention to the so-called "pitfalls" of AL evaluation proposed in [20]. For a complete list of the pitfalls and our considerations regarding them, please refer to App. B. To the best of our knowledge, we are the first to extend reliable SOTA (based on [13, 20]) experimentation to a total of 5 data domains and a high number of repetitions per experiment.

## 4    Few Repetitions are not Sufficient for Meaningful Results

To evaluate how many repetitions are necessary to obtain conclusive results in an AL experiment, we computed 100 runs of our top-performing AL method on one dataset. Our best method is margin sampling and we chose the Splice dataset for its average size and complexity.

This allows us firstly, to obtain a very strong estimation of the "true" average performance of margin sampling on this dataset and secondly, to draw subsets from this pool of 100 runs. Setting the size of our draws to $\alpha$ and sampling uniformly, we can approximate a cross-validation process with $\alpha$ repetitions. Each of these draws can be interpreted as a **reported result in AL literature** where the authors employed $\alpha$ repetitions. Figure 1 shows the "true" mean performance of margin sampling (green) in relation to random sampling (black) and the oracle performance (red). We display 5 random draws of size $\alpha$ in blue. We can observe that even for a relatively high number of repetitions the variance between the samples is extremely high, resulting in some performance curves being worse than random and some being significantly better. When setting $\alpha = 50$ we observe all samples to converge close to the true mean performance. In addition to this motivating example, we carried out our main evaluation (Tab. 3) multiple times by sampling 3 from our available runs uniformly at random and comparing the results. We found significant differences in the performance of AL methods on individual datasets, as well as permutations in the final ranking. This partly explains the ongoing difficulties in reproducing results for AL experiments and benchmarks. The details can be found in App. C. For this benchmark we employ 50 repetitions of every experiment.

### 4.1    Seeding vs. Repetitions

Considering the high computational cost of 50 repetitions, another approach to ensure consistency between experiments would be to reduce the amount of variance in the experiment by keeping as

many subsystems (weight initialization, data splits, etc.) as possible fixed with specialized seeding. We describe a novel seeding strategy in Appendix D that is capable of tightly controlling the amount variance in the experiment. However, previous works have noted that an actively sampled, labeled set does not generalize well between model architectures or even different initializations of the same model ([34, 19]), providing a bad approximation of the quality of an AL method (i.e. measured performances for an AL method might not even transfer to a different model initialization). Hence, we opt for letting the subsystems vary in controlled way (For details, please refer to App. D) and combine that with a high number of repetitions to obtain a good average of the generalization performance of each AL method.

# 5   CDALBench: A Cross-Domain Active Learning Benchmark

A detailed description of the preprocessing of each dataset can be found in Appendix E.

**Tabular:** AL research conducted on tabular data is sparse (only [1] from the considered baseline papers). We, therefore, introduce a set of tabular datasets that we selected according to the following criteria: (i) They should be solvable by medium-sized models in under 1000 samples, (ii) the gap between most AL methods and random sampling should be significant (potential for AL is present) and (iii) the gap between the AL methods and our oracle should also be significant (research on these datasets can produce further lifts). We use **Splice**, **DNA** and **USPS** from LibSVMTools [23].

Table 2: Employed model, chosen budget and available query sizes for each dataset

|  | Model | B | 1 | 5 | 20 | 50 | 100 | 500 | 1K |
|---|---|---|---|---|---|---|---|---|---|
| Semi DNA | Linear | 40 | o | o | | | | | |
| Semi Splice | Linear | 100 | o | o | o | | | | |
| TopV2 | BiLSTM | 200 | o | o | o | o | | | |
| Splice | MLP | 400 | o | o | o | o | o | | |
| DNA | MLP | 300 | o | o | o | o | o | | |
| USPS | MLP | 400 | o | o | o | o | o | | |
| Semi Cifar10 | Linear | 450 | o | o | o | o | o | | |
| Semi FMnist | Linear | 500 | o | o | o | o | o | | |
| Semi USPS | Linear | 600 | o | o | o | o | o | | |
| News | BiLSTM | 3K | | | o | o | o | o | |
| FMnist | ResNet18 | 10K | | | | | | o | o |
| Cifar10 | ResNet18 | 10K | | | | | | o | o |

**Image:** We use **FashionMNIST** [28] and **Cifar10** [16], since both are widely used in AL literature. **Text:** We use **News Category** [21] and **TopV2** [7]. Text datasets have seen less attention in AL research, but most of the papers that evaluate on text ([11], [34]) use at least one of these datasets. We use both, as they complement each other in size and complexity.

We would like to point out that these datasets are selected for speed of computation (both in terms of the required classifier and the necessary budget to solve the dataset). We are solely focused on comparing different AL methods in this paper and do not aim to develop novel classification models on these datasets. Our assumption is that a well-performing method in our benchmark will also generalize well to larger datasets and classifiers, because we included multiple different data domains, classifier types and sizes in our experiments.

Adapting the semi-supervised setting from [10], we offer all our datasets un-encoded (normal) as well as pre-encoded (semi-supervised) by a fixed embedding model that was trained by unsupervised contrastive learning . The text datasets are an exception to this, as they are only offered in their encoded form. Pre-encoded datasets enable us to test small query sizes on more complex datasets like Cifar10 and FashionMnist. They also serve the purpose of investigating the interplay between semi-supervised learning techniques and AL, as well as alleviating the cold-start problem described in [20] as they require a way smaller seed set. The classification model for every encoded dataset is a single linear layer with softmax activation. The embedding model was trained with the SimCLR [6] algorithm adopting the protocol from [10]. To ensure that enough information from the data is encoded by our embedding model, the quality of embeddings during pretext training was measured after each epoch. To this end, we attached a linear classification head to the encoder, fine-tuned it to the data and evaluated this classifier for test accuracy. The checkpoint of each encoder model will be provided together with the framework.

Every dataset has a fixed size for the seed set $\mathcal{L}^{(0)}$ of 1 sample per class, with the only exceptions being un-encoded FashionMnist and Cifar10 with 100 examples per class to alleviate the cold-start problem in these complex domains.

### 5.1 Query Sizes

We selected query sizes for each dataset to accommodate the widest range possible that results in a reasonable runtime for low query sizes and allows for at least 4 round of data acquisition for high query sizes. The available query sizes per dataset can be found in Table 2.

### 5.2 Realism vs. Variance

We would like to point out that some design choices for this framework prohibit direct transfer of our results to practical applications. This is a conscious choice, as we think that this is a necessary trade-off between realism and experiment variance. We would like to highlight the following design decisions:

(i) Creating test and validation splits from the full dataset rather than only the labeled seed set (following [20]). Fully fledged test and validation splits are unobtainable in practice, but they provide not only a better approximation of the methods generalization performance, but also a better foundation for hyperparameter tuning, which is bound to reduce variance in the experiment.

(ii) Choosing smaller classifiers instead of SOTA models. Since we are not interested in archiving a new SOTA in any classification problem, we instead opt to use smaller classifiers for the following reasons: Smaller classifiers generally exhibit more stable training behavior, on average require fewer sampled datapoints to reach their full-dataset-performance and have faster training times. For every dataset, the chosen architecture's hyperparameters are optimized to archive maximum full-dataset performance. Generally, we use MLPs for tabular, RestNet18 for image and BiLSTMs for text datasets. Every encoded dataset is classified by a single linear layer with softmax activation. The used model for each dataset can be found in Tab. 2. For a detailed description and employed hyperparameters please refer to Appendix E.

### 5.3 Evaluation Protocol

Following [34], the quality of an AL method is evaluated by an "anytime protocol" that incorporates classification performance at every iteration, as opposed to evaluating final performance after the budget is exhausted. We employ the normalized area under the accuracy curve (AUC):

$$\text{AUC}(\mathcal{D}_{\text{test}}, \hat{y}, B) := \frac{1}{B} \sum_{i=1}^{B} \text{Acc}(\mathcal{D}_{\text{test}}, \hat{y}^{(i)}) \tag{1}$$

Since AUC is still influenced by the budget, we define a set of rules to set this hyperparameter upfront, so that we are not favoring a subset of methods by handcrafting a budget. In this work, we choose the budget per dataset to be the first point at which one of 2 stopping conditions apply: (i) an method (except oracle) manages to reach 99% of the full-dataset-performance (using the smallest query size) or (ii) the best method (except oracle) did not improve the classifier's accuracy by at least 2% in the last 20% of iterations. The first rule follows [13], while the second rule prevents excessive budgets for cases with diminishing returns in the budget. The resulting budgets can be found in Tab. 2.

As described in Sec. 4, we repeat each experiment 50 times. Each repetition retains the train/test split (often given by the dataset itself), but creates a new validation split that is sampled from the entire dataset (not just the seed set $\mathcal{L}^{(0)}$).

Apart from plotting standard performance curves and reporting their AUC values per dataset in App. F, we primarily rely on ranks to aggregate the performance of an AL method across datasets. For each dataset and query size, the AUC values of all AL methods are sorted and assigned a rank based on position, with the best rank being 1. These ranks can safely be averages across datasets as they are no longer subjected to scaling differences of each dataset. Additionally, we employ Critical Difference (CD) diagrams (like Fig. 2) for statistical testing. CD diagrams [12] use the Wilcoxon signed-rank test, which is a variant of the paired T-test, to find significant differences of ranks between AL methods. For a detailed description of how every CD diagram is created, please refer to App. G.

# 6 A Greedy Oracle Algorithm

Using additional resources, like excessive training time, or direct access to a labeled test set, an oracle method for AL finds the oracle set $\mathcal{O}_b$ for a given dataset, model, and training procedure that induces the highest AUC score for a given budget. However, due to the combinatorial nature of the problem, this is computationally infeasible for realistic datasets. Hence, previous works have proposed approximations to this oracle sequence. [34] used simulated annealing to search for a subset with maximal test accuracy and used the best solution after a fixed time budget. Even though their reported performance curves display a significant lift over all other AL methods, we found the computational cost of reproducing this oracle for all our datasets to be prohibitive (The authors reported the search to take several days per dataset on 8 V100 GPUs). In this paper, we propose a greedy oracle algorithm that constructs an approximation of the optimal set in an iterative fashion. Our oracle algorithm uniformly samples at iteration $i$ a subset $\mathcal{U}_S$ of size $\tau$ of the not already labeled data points $\mathcal{U}^{(i)}$. Then it recovers the label $y$ for each of the sampled $u \in \mathcal{U}_S$ and selects the point $u$ for which the classifier $\hat{y}^{(i)}$ trained on $\mathcal{L}^{(i)} \cup \{u\}$ has maximal performance. Due to the algorithms greedy nature (considering only the next point to pick), our oracle frequently encounters situations where every point in $u$ would incur a negative lift (worsening the test performance). This can happen, for example, if the oracle picked a labeled set that enables the classifier to correctly classify a big portion of easy samples in the test set, but now fails to find the next **single** unlabeled point that would enable the classifier to succeed on one of the hard samples. This leads to a situation, where no point can immediately incur an increase in test performance and therefore the selected data point can be considered random. To circumvent this problem, we use our best-performing AL method (margin sampling [27]) as a fallback option for the oracle. Whenever the oracle does not find an unlabeled point that results in an increase in performance, it defaults to margin sampling from the entire unlabeled pool $\mathcal{U}^{(i)}$ in that iteration. The resulting greedy algorithm constructs an approximation of the optimal labeled set that consistently outperforms all other algorithms by a significant margin, while requiring relatively low computational cost ($\mathcal{O}(B\tau)$). We fix $\tau = 20$ in this work, as this gives us an average lift of 5% over the best performing AL method per dataset (which is significant for AL settings) and we expect diminishing returns for larger $\tau$. The pseudocode for our oracle can be found in App. H. Even though our proposed algorithm is more efficient than other approaches, the computational costs for high budget datasets like Cifar10 and FashionMnist meant that we could not compute the oracle for all 10000 datapoints. To still provide an oracle for these two datasets, we select two points per iteration instead of one and stop the oracle computation at a budget of 2000. The rest of the curve is forecast with a 2-stage linear regression that asymptotically approaches the upper bound performance of the dataset. A detailed description can be found in App. I.

# 7 Experiments

## 7.1 Implementation Details

At each iteration $i$ the AL method picks an unlabeled datapoint based on a fixed set of information $\{\mathcal{L}^{(i)}, \mathcal{U}^{(i)}, B, |\mathcal{L}^{(i)}| - |\mathcal{L}^{(1)}|, \text{acc}^{(i)}, \text{acc}^{(1)}, \hat{y}^{(i)}, \text{opt}_{\hat{y}}\}$, where $\text{opt}_{\hat{y}}$ is the optimizer used to fit $\hat{y}^{(i)}$. This set grants full access to the labeled and unlabeled set, as well as all parameters of the classifier and the optimizer. Additionally, we provide meta-information, like the size of the seed set through $|\mathcal{L}^{(i)}| - |\mathcal{L}^{(1)}|$, the remaining budget though the addition of $B$ and the classifiers potential through $\text{acc}^{(1)}$ and $\text{acc}^{(i)}$. We allow AL methods to derive information from this set, e.g. predictions of the classifier $\hat{y}^{(i)}(x); \ x \in \mathcal{U}^{(i)} \cup \mathcal{L}^{(i)}$, clustering, or even training additional models. However, the method may not incorporate external information e.g. other datasets, queries to recover additional labels, additional training steps for $\hat{y}$, or the test/validation set.

For our study we selected AL methods with good performances reported by multiple different sources that can work with the set of information stated above. For a list of all AL methods, please refer to Table 3, with detailed descriptions being found in Appendix J.

The model $\hat{y}^{(i)}$ can be trained in two ways. Either the parameters of the model are reset to a fixed initial setting $\hat{y}^{(0)}$ after each AL iteration and the classifier is trained from scratch with the updated labeled set $\mathcal{L}^{(i)}$, or the previous state $\hat{y}^{(i-1)}$ is retained and the classifier is fine-tuned on $\mathcal{L}^{(i)}$ for a reduced number of epochs. In this work, we use the fine-tuning method for un-encoded datasets to

Table 3: Performances for AL methods on real-world datasets, aggregated for un-encoded (normal) and encoded (semi-supervised) datasets. Performance is shown as average ranks over repetitions (1.0 is the best rank). Methods are sorted by aggregated performance on un-encoded (normal) datasets.

|  | Splice | DNA | USPS | Cifar10 | FMnist | TopV2 | News | Normal | Semi |
|---|---|---|---|---|---|---|---|---|---|
| Oracle | 1.0±0.01 | 1.0±0.01 | 1.0±0.0 | 1.0±0.0 | 1.0±0.0 | 1.0±0.01 | 1.0±0.0 | 1.0 | 2.1 |
| Margin | 6.8±0.02 | 4.5±0.01 | 2.7±0.01 | 7.2±0.01 | 4.9±0.0 | 3.1±0.01 | 4.5±0.0 | 4.8 | 4.7 |
| Galaxy | 9.5±0.02 | 9.4±0.02 | 2.4±0.01 | 2.7±0.01 | 5.6±0.01 | 2.6±0.01 | 2.0±0.0 | 4.9 | 5.7 |
| Badge | 6.2±0.01 | 6.5±0.01 | 3.8±0.01 | 6.2±0.01 | 5.1±0.0 | 4.2±0.01 | 3.6±0.0 | 5.1 | 6.0 |
| LeastConfident | 9.6±0.02 | 11.1±0.02 | 9.1±0.02 | 2.6±0.01 | 4.4±0.0 | 8.9±0.02 | 5.2±0.01 | 7.3 | 7.1 |
| DSA | 7.8±0.02 | 7.7±0.01 | 8.5±0.01 | 6.4±0.01 | 5.6±0.0 | 7.0±0.02 | 8.1±0.01 | 7.3 | 7.3 |
| CoreGCN | 7.2±0.01 | 5.2±0.01 | 11.4±0.01 | 8.6±0.01 | 7.1±0.01 | 5.0±0.01 | 7.4±0.01 | 7.4 | 8.9 |
| BALD | 4.1±0.01 | 4.8±0.01 | 6.4±0.01 | 13.0±0.01 | 8.4±0.01 | 8.6±0.02 | 7.7±0.0 | 7.6 | 8.3 |
| Entropy | 6.8±0.02 | 4.0±0.01 | 8.6±0.01 | 8.6±0.01 | 5.4±0.01 | 10.8±0.02 | 11.1±0.01 | 7.9 | 7.3 |
| LSA | 6.2±0.01 | 7.2±0.01 | 6.3±0.01 | 8.6±0.01 | 11.6±0.01 | 8.5±0.01 | 7.7±0.0 | 8.0 | 8.1 |
| Random | 9.4±0.01 | 9.7±0.01 | 6.3±0.01 | 9.4±0.01 | 12.1±0.0 | 8.9±0.01 | 8.3±0.0 | 9.2 | 7.6 |
| Coreset | 7.3±0.01 | 9.5±0.01 | 11.5±0.01 | 7.7±0.01 | 7.8±0.0 | 9.5±0.02 | 11.5±0.01 | 9.3 | 7.9 |
| TypiClust | 9.1±0.01 | 10.6±0.01 | 13.0±0.02 | 8.9±0.01 | 12.0±0.01 | 13.0±0.02 | 13.0±0.01 | 11.4 | 9.9 |

Figure 2: Ranks of each AL method aggregated by domain. Horizontal bars indicate a **non**-significant rank difference. The significance is tested via a paired-t-test with $\alpha = 0.05$.

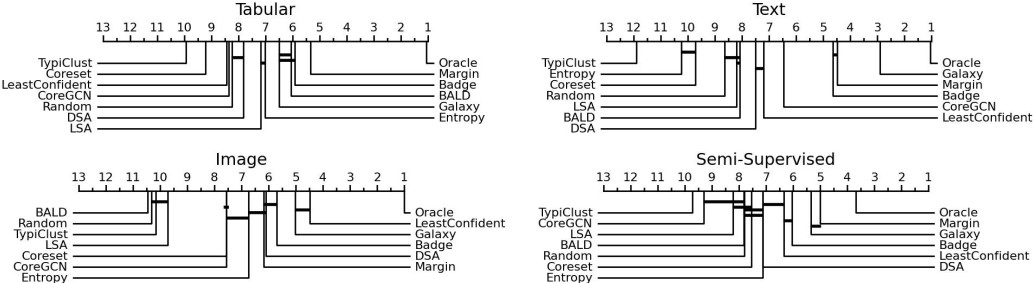

save computational time, while we use the from-scratch training for encoded datasets since they have very small classifiers and this approach generally produces better results. Our fine-tuning scheme always trains for at least one epoch and employs an aggressive early stopping with a patience of 2 afterwards.

## 7.2 Results on Real-world Data

In Table 3 we provide the rank of each AL method per dataset. Please note, that we are averaging not only over runs, but also over query sizes per dataset, impacting AL methods that do not adapt well to a wide range of query sizes. For the results per query size, please refer to App. K. As mentioned in contribution 3, our results on real-world data show significant differences in the performance of methods between data domains: Not only do some methods overperform on some domains (like least confidence (LC) sampling on images), but the Top-3 of methods (except oracle) does not contain the same three methods for **any** two domains. Most interestingly, the image domain, which received most of the attention in benchmarking so far could even be considered an outlier, as this is the only domain where the Top-1 method changes. This highlights the dire need for diverse data domains in AL benchmarking.

Results for the semi-supervised domain appear mostly in line with the other 3 domains, but a closer analysis of performances split into encoded images and encoded tabular reveals the need for further research. For details, please refer to App. L.

Finally, we would like to emphasize that the total average rank of our top 3 algorithms (column "Normal" in Tab. 3) are 4.8, 4.9 and 5.1. No single algorithm was able to perform well in every domain, either being outperformed by a specialist algorithm in each domain, or experiencing a severe drop in performance in a poorly matched domain.

## 8 Honeypot and Diverging Sine

AL approaches can be categorized into two types: uncertainty and geometric approaches. Typical members of the first category are variants of uncertainty sampling like entropy, margin and LC sampling [27] as well as BALD [9]. Typical members of the second category are clustering approaches

Figure 3: Synthetic "Honeypot" and "Diverging Sine" datasets. The optimal decision boundary is not part of the dataset and serves only as a visual guide.

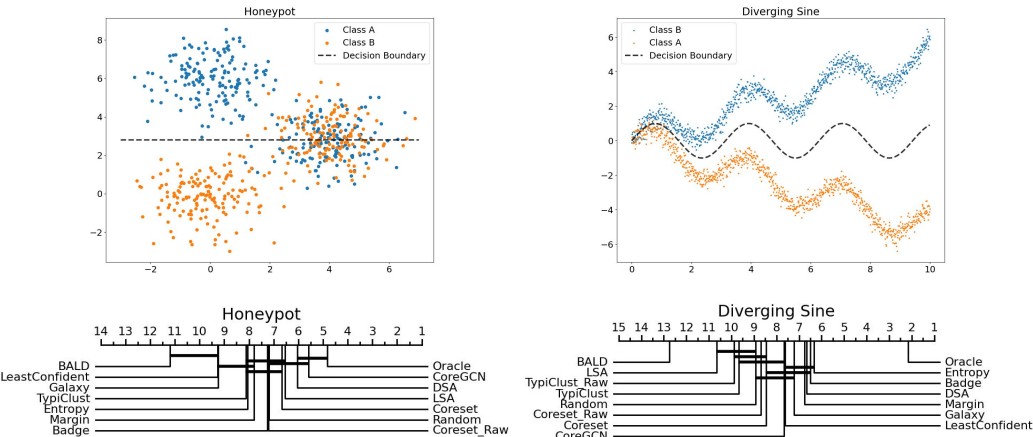

like Coreset [26], BADGE [1] and TypiClust [10]. Both types of methods have principled shortcomings in terms of their utilized information that makes them unsuitable for certain data distributions. To test for these specific shortcomings, we created two synthetic datasets, namely "Honeypot" and "Diverging Sine", that are hard to solve for methods focused on the classifier's decision boundary or data clustering respectively. To avoid methods memorizing these datasets, they are generated from scratch for each experiment.

Honeypot creates two easy to distinguish clusters and one "honeypot" that represents a noisy region of the dataset with potentially miss-labeled, miss-measured or generally adverse samples. The honeypot is located on the likely decision boundary of a classifier that is trained on the beneficial samples to maximize its adverse impact on purely uncertainty-based AL methods. Diverging Sine samples datapoints for each class from two diverging sinusoidal functions that are originating from the same y-intercept. This creates a challenging region on the left hand side, where a lot of datapoints need to be sampled, and an easy region on the right hand side, where very few datapoints are sufficient. The repeating nature of a sine function encourages diversity-based AL methods to equally sample the entire length, drastically oversampling the right hand side of the dataset.
Both datasets have a budget of $B = 60$ and are tested with query sizes 1 and 5.

We provide the rank of all AL Methods on Honeypot and Diverging Sine in Fig. 3. Results for the Honeypot dataset reveal expected shortcomings of uncertainty sampling methods like margin, entropy and LC sampling as well as BALD. In addition, BADGE is underperforming for this dataset compared to real-world data. Both margin sampling and BADGE (the two best methods) being vulnerable to adverse samples or simply measurement noise, highlights the need for further research into robust AL methods.
Results for Diverging Sine also confirm expected behavior, as clustering methods (Coreset, TypiClust) fall behind uncertainty methods (entropy, margin, LC sampling), with the exception of BADGE. The fact that BADGE is able to perform well on Diverging Sine highlights the importance of embeddings for the clustering methods, as the gradient embedding from BADGE seems to be able to encode uncertainty information, guiding the selection into the left hand regions of the dataset. We provide a small ablation study on the importance of the embeddings by testing a version of Coreset and TypiClust on this dataset that does not use the embeddings produced by the classification model, but rather clusters the data directly. "Coreset Raw" and "TypiClust Raw" both perform worse than their embedding-based counterpart.

## 9    Comparison to other Benchmarks

Comparing our results to the findings of other works based in accuracy scores would be meaningless, as every work employs different models, hyperparameters and training loops. We instead opt to compare only the ranking of algorithms to the literature.
Our results generally are reflected in domain-specific benchmarks - [32] also find least confidence sampling and BADGE to be the best algorithms for images (they don't test Galaxy), [25] also find

BADGE to be the best algorithm for text (they don't test margin sampling or Galaxy) and [2] also find margin sampling to be the best algorithm for tabular data.

In this work, we provide the first AL benchmark that spans all major data domains and is easily reproducible while matching and extending previous published results from single-domain benchmarks.

## 10    Using this Benchmark

We strongly advocate to test newly proposed AL methods not only on a wide variety of real data domains, but also to pay close attention to the Honeypot and Diverging Sine datasets to reveal principled shortcomings of the method in question. Both tasks can be easily carried out by implementing the new AL method into our code base. For Limitations and Future Work, please refer to App. O.

**Acknowledgement**    Funded by the Lower Saxony Ministry of Science and Culture under grant number ZN3492 within the Lower Saxony "Vorab" of the Volkswagen Foundation and supported by the Center for Digital Innovations (ZDIN).

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

# A   AL Recommendations from Ji et al.

**Recommendation 1**   Use the backbone architecture with the community-accepted definition that is best suited for the dataset at hand and consistently use it across all experiments. In the image classification domain, we suggest using ResNet18 for CIFAR-10 and CIFAR-100.
→ We are using Resnet18 for our image datasets and re-purposed the LSTM model from [34] for the text datasets. For the tabular data, we ran a grid-search (full dataset test accuracy) over MLP architectures.

**Recommendation 2**   Control the type of optimizer across methods for comparative evaluations to ensure that the yield performance difference stems from an active learning method itself. As SGD often generalizes better, we encourage its use for deep active learning.
→ We searched for the best optimizer per dataset via generalization performance (test accuracy).

**Recommendation 3**   Pragmatically fix the learning rate to 0.1 for SGD on image datasets. While continuous hyperparameter tuning can improve overall performance, a fixed learning rate does not change the ranking of AL methods from a comparative evaluation's point of view.
→ We found that for smaller datasets, a learning rate of 0.1 was unsuitable. Ji et al. only used large image datasets, so a learning rate of 0.1 was sufficient for them. The learning rate is part of our hyperparameter grid-search.

**Recommendation 4**   One may use data augmentation if applied consistently across methods, such that it does not affect the overall ranking. However, a commonly accepted baseline is needed, e.g., random horizontal flipping and random cropping for image classification.
→ We did not find a data augmentation technique that could be applied equally on all datasets, so we refrained from it. The only possibility would be Gaussian noise, but the impact of gaussian noise on pre-encoded is not well-understood.

**Recommendation 5**   Refine model parameters across AL batch ("warm starts") to prevent exhaustive reinitialization and feed initialization of the backbone model's weights and the "init sets" with fixed inputs over multiple runs to average out the randomness. Moreover, use identical seeds for all methods under investigation.
→ We employed a novel seeding strategy to closely control the seeding of our experiments (Details in App. D). We applied warm-starts for most datasets, except pre-encoded ones, because we found a generally better performance, if we train the classifier from scratch.

**Recommendation 6**   Run experiments multiple times to compensate for non-deterministic operations. If the resulting variance is larger than the gained improvement, use deterministic operations stringently
→ A study on (non-)deterministic operations has not been conducted in this work, but our large number of repetitions compensate for that.

**Recommendation 7SW**   Configure and verify influence parameter in active learning implementations thoroughly. To foster future research, we provide implementations as part of our framework at: https://intellisec.de/research/eval-al
→ We provide our own baseline code, since we implemented a novel seeding strategy and unify many additional data domains in the code.

**Recommendation 7HW**   Ensure that comparative evaluations are run on identical hardware. While it is not necessary to execute all experiments on the same physical device, the GPU model, for instance, should be the same. Do not mix hardware and list hardware details.
→ The large computational cost of our benchmark did not allow us to compute on only one type of hardware (Which would mean to only use part of our cluster). However, our large number of repetitions compensate for that.

**Recommendation 8**   Consider multiple query-batch sizes in the evaluation. The choice of the sizes needs to be appropriate for the total number of unlabeled samples.
→ We employed a wide range of batch sizes. For details, please refer to Table 2.

**Recommendation 9** Compare active learning strategies without sub-sampling, unless one of the approaches uses it as a fundamental building block. In this case a detailed analysis of the influence of sub-sampling is necessary.
→ We only carefully employed sub-sampling, when it was absolutely necessary to keep the computation times feasible.

**Recommendation 10** Evaluate active learning strategies on multiple benchmark datasets, that comprise balanced, imbalanced, small-scale, and large-scale datasets to cover most relevant cases in practice.
→ We deferred the study on (im-)balanced datasets to future work. However, our benchmark contains datasets of many different sizes and we extend this argument to domains as well.

**Recommendation 11** For a comprehensive analysis of AL strategies, the overall comparative evaluation should incorporate as many variables from Section 3 to yield a summarized PPM that is as expressive as possible.
→ In our main result (Tab. 3) we average the performance over runs, query sizes and datasets.

# B    AL Pitfalls from Lueth et al.

**P1 Data distribution** The proposed evaluation over a diverse selection of dataset distributions including specific roll-out datasets proved essential for realistic evaluation of QMs as well as the different training strategies. One main insight is the fact that class distribution is a crucial predictor for the potential performance gains of AL on a dataset: Performance gains of AL are generally higher on imbalanced datasets and occur consistently even for ST models with a small starting budget, which are typically prone to experience cold start problems. This observation is consistent with a few previous studies [...].
→ We selected our datasets according to their "potential" for AL. We measured this potential by the distance of most AL methods to random and the distance of the best AL method to our oracle. If both distances are ¿0, we consider the dataset useful.

**P2 Starting budget** The comprehensive study of various starting budgets on all datasets reveals that AL methods are more robust with regard to small starting budgets than previously reported [6, 20, 43]. With the exception of Entropy we did not observe cold start problems even for any QM even in combination with notoriously prone ST models. The described robustness is presumably enabled by our thorough classifier configuration (P4) and heuristically adapted query sizes (P3). This finding has great impact potential suggesting that AL can be applied at earlier points in the annotation process thereby further reducing the labeling cost [...].
→ Our experiments also showed a high resistance against the cold-start problem, which prompted us to use the smallest possible seed set for most datasets (1 point per class), with the only exception being (un-)encoded Cifar10 and FashionMnist. Here we employ a seed set of 100 points per class to avoid a cold-start.

**P3 Query size** Based on our evaluation of the query size we can empirically confirm its importance with regard to 1) general AL performance and 2) counteracting the cold start problem. The are, however, surprising findings indicating that the exact interaction between query size and performance remains an open research question. [...].
→ We generally observed a decreasing performance for larger query sizes. We therefore made sure, that we include the smallest possible query sizes that result in feasible computation times.

**P4 Classifer configuration** Our results show that method configuration on a properly sized validation set is essential for realistic evaluation in AL. [...] This raises the question of to which extent reported AL advantages could have been achieved by simple classifier configurations. Further, our models also generally outperform expensively configured models by Munjal et al.. Thus, we conclude that manually constraining the search space renders HP optimization feasible in practice without decreasing performance and ensures performance gains by Active Learning are not overstated. The importance of the proposed strategy to optimize HPs on the starting budget for each new dataset is supported by the fact that the resulting configurations change across datasets.

→ We also strongly advocate the use of a fully fledged validation set for HP tuning, as this allows for a higher quality of HPs, which in turn reduces the variance of the experiments.

**P5 Alternative training paradigms** Based on our study benchmarking AL in the context of both Self-SL and Semi-SL, we see that while Self-SL generally leads to improvements across all experiments, Semi-SL only leads to considerably improved performance on the simpler datasets CIFAR-10/100, on which Semi-SL methods are typically developed. Generally, models trained with either of the two training paradigms receive a lower performance gain from AL (over random querying) compared to ST. [...] The fact that AL entails multiple training iterations amplifies the computational burden of Semi-SL, rendering their combination prohibitively expensive in most practical scenarios. Further, the fact that our Semi-SL models based on Fixmatch do not seem to generalize to more complex datasets in our setting stands in stark contrast to conclusions drawn by [...] as to which the emergence of Semi-SL renders AL redundant. Interestingly, the exact settings where Semi-SL does not provide benefits in our study are the ones where AL proved advantageous. The described contradiction with the literature underlines the importance of our proposed protocol testing for a method's generalizability to unseen datasets.
→ Due to the high computational costs described by Lueth et al., we opted for the most efficient form of semi-supervised learning, which is to train a fixed encoder-model, pre-encode the datasets and then only train a single linear layer as classifier.

## C  Difference of Ranks with 3 Repetitions

Table 4 and Table 5 follow the exact same computation of ranks that created the main result (Table 3) with the only difference being a reduced number of runs per AL method. For each table we sampled 3 runs uniformly at random from the available 50 per AL method.
We can observe significant differences between the two tables:
Purple: A multitude of rank differences of AL methods for specific datasets, some as high as 4.7 ranks for TypiClust on the Splice dataset
Olive: Well separated AL methods in Tab. 5 (Margin and BADGE) are almost indistinguishable in Tab 4
Red: BALD lost 2 places in the overall ranking and Entropy gained 2

Even though the overall ordering of AL methods stayed relatively unchanged due to the averaging across many datasets, each individual dataset was subject to drastic permutations. This highlights the need for many repetitions in AL experiments.

Table 4: Ranks of all AL methods per dataset. First random draw of 3 runs from the overall pool of 50.

|          | Splice | DNA  | USPS | Cifar10 | FMnist | TopV2 | News | Unencoded | Encoded |
|----------|--------|------|------|---------|--------|-------|------|-----------|---------|
| Oracle   | 1.0    | 1.0  | 1.0  | 1.0     | 1.0    | 1.0   | 1.0  | 1.0       | 2.1     |
| Margin   | 6.0    | 7.3  | 2.0  | 6.7     | 5.3    | 2.3   | 3.3  | 4.7       | 4.4     |
| Badge    | 6.0    | 7.3  | 3.0  | 6.7     | 5.0    | 3.3   | 4.0  | 5.0       | 5.3     |
| BALD     | 3.3    | 4.7  | 5.3  | 12.0    | 7.0    | 6.3   | 4.3  | 6.1       | 7.9     |
| CoreGCN  | 8.7    | 3.7  | 10.7 | 6.3     | 5.3    | 4.0   | 7.7  | 6.6       | 9.1     |
| DSA      | 8.3    | 6.3  | 7.7  | 7.7     | 4.3    | 6.7   | 6.7  | 6.8       | 6.1     |
| LeastConf| 10.0   | 12.0 | 8.0  | 3.0     | 4.3    | 9.3   | 2.3  | 7.0       | 6.7     |
| LSA      | 5.7    | 6.7  | 5.3  | 6.7     | 10.7   | 7.7   | 7.0  | 7.1       | 6.3     |
| Entropy  | 11.0   | 3.3  | 7.3  | 4.0     | 6.7    | 8.3   | 9.7  | 7.2       | 7.0     |
| Random   | 7.7    | 8.7  | 5.3  | 8.0     | 11.0   | 8.0   | 9.0  | 8.2       | 6.3     |
| Coreset  | 4.7    | 10.3 | 10.3 | 7.7     | 6.0    | 9.0   | 11.0 | 8.4       | 7.2     |
| TypiClust| 5.7    | 6.7  | 12.0 | 8.3     | 11.3   | 12.0  | 12.0 | 9.7       | 9.7     |

Table 5: Ranks of all AL methods per dataset. Second random draw of 3 runs from the overall pool of 50.

| | Splice | DNA | USPS | Cifar10 | FMnist | TopV2 | News | Unencoded | Encoded |
|---|---|---|---|---|---|---|---|---|---|
| Oracle | 1.0 | 1.0 | 1.0 | 1.0 | 1.0 | 1.0 | 1.0 | 1.0 | 2.4 |
| Margin | 6.0 | 3.3 | 2.0 | 5.7 | 2.0 | 2.0 | 4.3 | 3.6 | 3.8 |
| Badge | 6.0 | 9.0 | 3.0 | 3.0 | 5.7 | 3.7 | 3.3 | 4.8 | 4.9 |
| CoreGCN | 4.3 | 6.3 | 10.3 | 7.3 | 5.3 | 5.7 | 5.3 | 6.4 | 8.1 |
| DSA | 8.7 | 7.3 | 7.3 | 6.0 | 4.3 | 5.3 | 6.0 | 6.4 | 6.5 |
| BALD | 4.7 | 4.0 | 4.7 | 12.0 | 7.3 | 6.7 | 6.7 | 6.6 | 7.5 |
| Entropy | 6.7 | 4.7 | 7.7 | 5.3 | 5.0 | 7.3 | 9.3 | 6.6 | 6.8 |
| LeastConf | 7.7 | 10.0 | 8.3 | 3.3 | 6.0 | 8.7 | 3.0 | 6.7 | 7.3 |
| LSA | 7.7 | 5.3 | 6.0 | 9.0 | 11.0 | 9.0 | 7.3 | 7.9 | 7.5 |
| Random | 9.3 | 8.0 | 5.0 | 8.7 | 11.7 | 8.3 | 8.7 | 8.5 | 7.6 |
| Coreset | 6.0 | 10.7 | 10.7 | 8.0 | 8.3 | 8.3 | 11.0 | 9.0 | 6.3 |
| TypiClust | 10.0 | 8.3 | 12.0 | 8.7 | 10.3 | 12.0 | 12.0 | 10.5 | 9.4 |

# D    Seeding Strategy

We aim to provide an experimental setup that is fully reproducible independent of the dataset, classification model, or AL method used. For a fair comparison of two AL methods, both methods need to receive equal starting conditions in terms of train/validation split, initialization of classifier, and even the state of minor systems like the optimizer or mini-batch sampler. Even though different implementations might have their own solution to some of these problems, only [13] has described and implemented a fully reproducible pipeline for AL evaluation. The term reproducibility in this work is used as a synonym not only for the reproducibility of an experiment (a final result given a seed), but also the reproducibility of all subsystems independent of each other. The seed for one subsystem should always reproduce the behavior of this subsystem independent of all other subsystems and their seeds. The main obstacle for ensuring reproducibility is the seeding utility in PyTorch, Tensorflow, and other frameworks, whose default choice is a single global seed. Since many subsystems draw random numbers from this seed, all of them influence each other to a point where a single additional draw can completely change the model initialization, data split or the order of training batches. Even though some workarounds exist, e.g. re-setting the seed multiple times, this problem is not limited to the initialization phase, but also extends to the AL iterations and the systems within. We propose an implementation that creates separate Random Number Generators (RNGs) for each of these systems to ensure equal testing conditions even when the AL method, dataset, or classifier changes. We hypothesize that the insufficient setup with global seeds contributes to the ongoing problem of inconsistent results of AL methods in different papers.

In summary, we introduce three different seeds: $s_\Omega$ for the AL method, $s_\mathcal{D}$ for dataset splitting and mini-batch sampling, and $s_\theta$ for model initialization and sampling of dropout masks. Unless stated otherwise, we will keep $s_\Omega$ fixed, while $s_\mathcal{D}$ and $s_\theta$ are incremented by 1 between repetitions to introduce stochasticity into our framework. Some methods require a subsample to be drawn from $\mathcal{U}$ in order to reduce the computational cost in each iteration, while others need access to the full unlabeled pool (e.g. for effective clustering). If a subsample is required, it will be drawn from $s_\Omega$ and therefore will not influence other systems in the experiments. For each method, we decided if subsampling is required based on our available hardware, but decided against setting a fixed time limit per experiment, since this would introduce unnecessary complexity into the benchmark. An overview of selected hyperparameters per AL method can be found in Appendix M.
**Note:** Even though we decoupled the subsystems via the described seeds, the subsystems can still influence each other in a practical sense. For example, keeping $s_\mathcal{D}$ fixed does not mean that always the same sequence of samples from $\mathcal{U}$ (if subsamples are drawn) are shown to all AL methods. This is practically impossible, as different AL methods pick different $x^{(i)}$. However, the hypothetical **tree** of all possible sequences of samples from $\mathcal{U}$ remains the same, granting every AL methods equal possibilities.

# E    Hyperparameters and Preprocessing per Dataset

For all our datasets we use the pre-defined train/test splits, if given. In the remaining cases, we define test sets upfront and store them into separate files to keep them fixed across all experiments.

The validation set is split in the experiment run itself and depends on the dataset-seed.

**Tabular:** We use **Splice**, **DNA** and **USPS** from LibSVMTools [23]. All three datasets are normalized between [0, 1].

**Image:** We use **FashionMNIST** [28] and **Cifar10** [16], since both are widely used in AL literature. Both datasets are normalized according to their standard protocols.

**Text:** We use **News Category** [21] and **TopV2** [7]. For News Category we use the 15 most common categories as indicated by its Kaggle site. We additionally drop sentences above 80 words to reduce the padding needed (retaining 99,86% of the data). For TopV2, we are only using the "alarm" domain. Both datasets are encoded with pre-trained GloVe (Common Crawl 840B Tokens) embeddings [24]. Since neither dataset provided a fixed test set, we randomly split 7000 datapoints into a test set.

| Dataset | Seed Set | Budget | Val Split |
|---|---|---|---|
| Splice | 1 | 400 | 0.2 |
| SpliceEnc. | 1 | 60 | 0.2 |
| DNA | 1 | 300 | 0.2 |
| DNAEnc | 1 | 40 | 0.2 |
| USPS | 1 | 400 | 0.2 |
| USPSEnc | 1 | 600 | 0.2 |
| FashionMnist | 100 | 2000 | 0.04 |
| FashionMnistEnc | 1 | 500 | 0.04 |
| Cifar10 | 100 | 2000 | 0.04 |
| Cifar10Enc | 1 | 350 | 0.04 |
| TopV2 | 1 | 125 | 0.25 |
| News | 1 | 1500 | 0.03 |

Table 6: Size of the seed set is given by number of labeled sample per class.

| Dataset | Classifier | Optimizer | LR | Weight Decay | Dropout | Batch Size |
|---|---|---|---|---|---|---|
| Splice | [24, 12] | NAdam | 1.2e-3 | 5.9e-5 | 0 | 43 |
| SpliceEnc. | linear | NAdam | 6.2e-4 | 5.9e-6 | 0 | 64 |
| DNA | [24, 12] | NAdam | 3.9e-2 | 3.6e-5 | 0 | 64 |
| DNAEnc | linear | NAdam | 1.6e-3 | 4e-4 | 0 | 64 |
| USPS | [24, 12] | Adam | 8.1e-3 | 1.5e-6 | 0 | 43 |
| USPSEnc | linear | NAdam | 7.8e-3 | 1.9e-6 | 0 | 64 |
| FashionMnist | ResNet18 | NAdam | 1e-3 | 0 | 0 | 64 |
| FashionMnistEnc | linear | Adam | 1.6e-3 | 1e-5 | 5e-2 | 64 |
| Cifar10 | ResNet18 | NAdam | 1e-3 | 0 | 0 | 64 |
| Cifar10Enc | linear | NAdam | 1.7e-3 | 2.3e-5 | 0 | 64 |
| TopV2 | BiLSTM | NAdam | 1.5e-3 | 1.7e-7 | 5e-2 | 64 |
| News | BiLSTM | NAdam | 1.5e-3 | 1.7e-7 | 5e-2 | 64 |

Table 7: Classifier architectures and optimized hyperparameters per dataset. Numbers in brackets signify a MLP with corresponding hidden layers.

# F  Performance Curves per Dataset

Figure 4: Performance curves per query size for normal (un-encoded) Splice

Figure 5: Performance curves per query size for semi-supervised (encoded) Splice

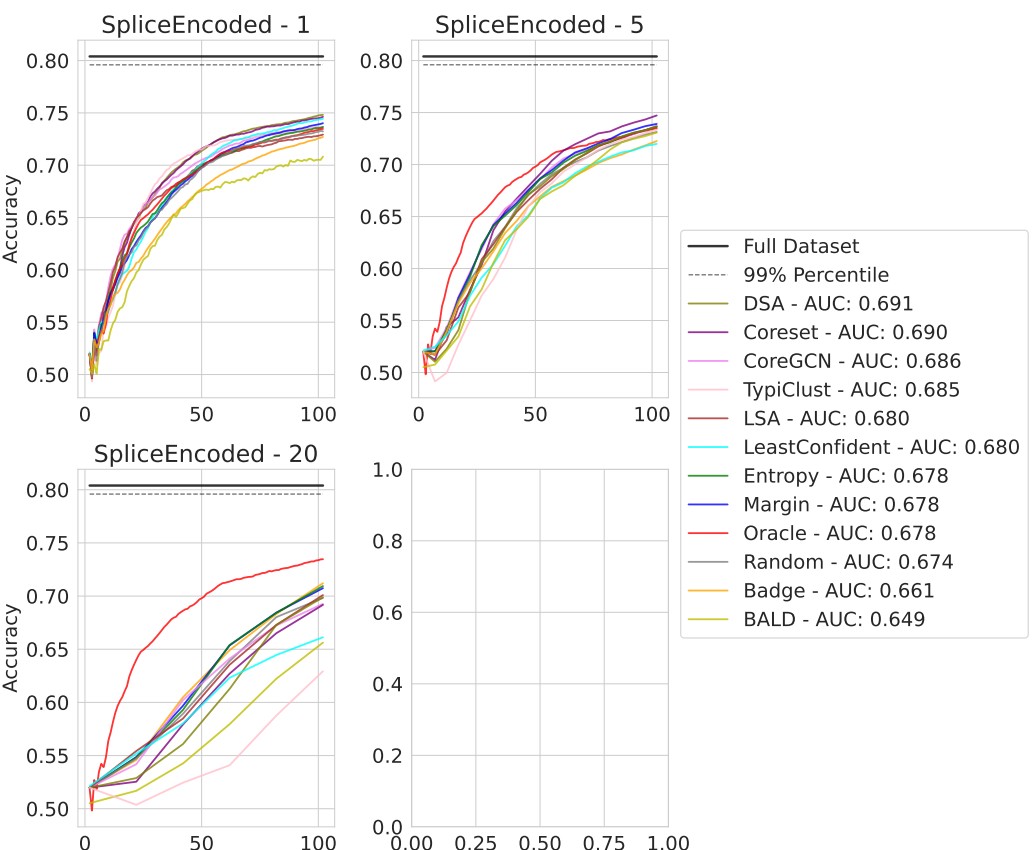

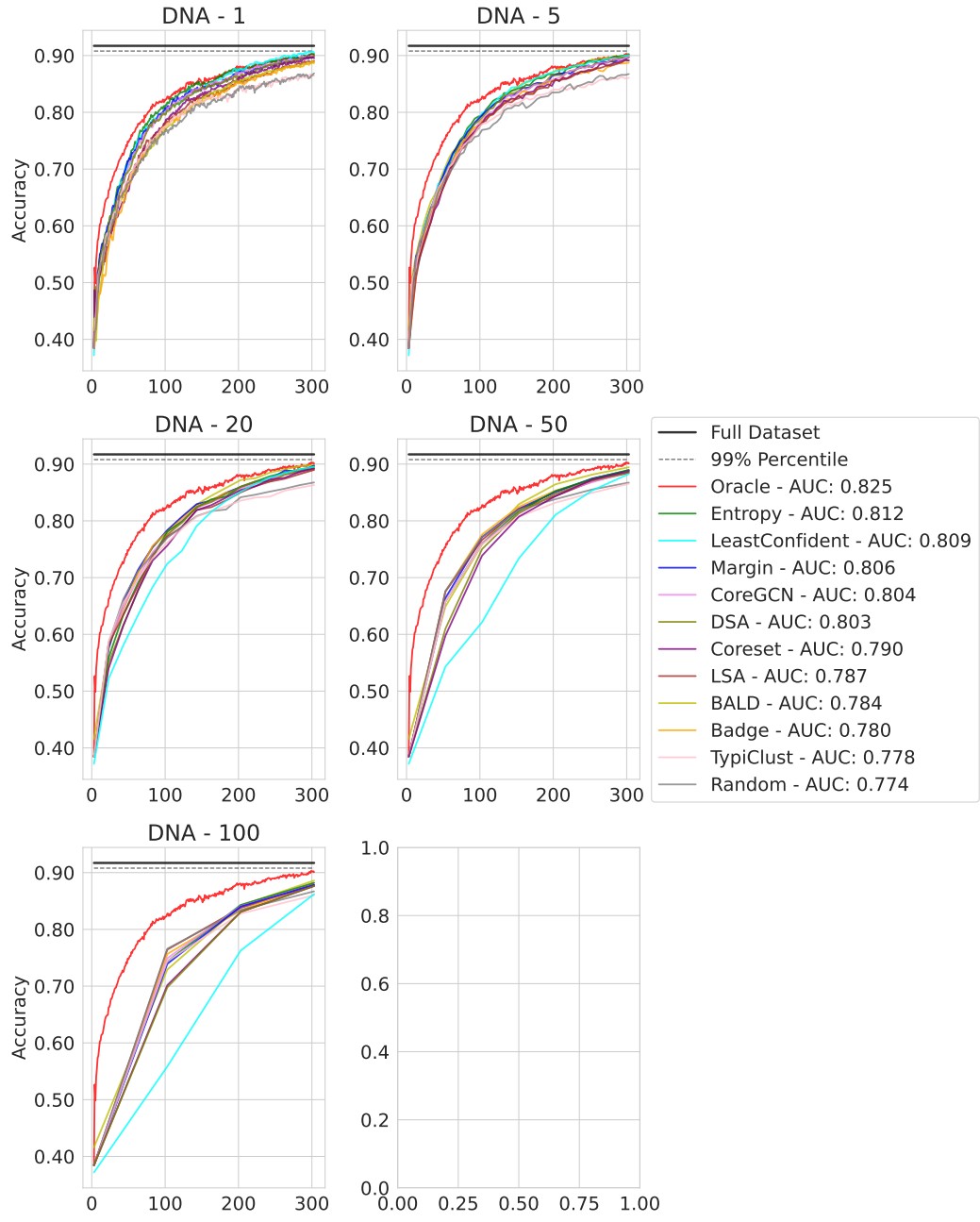

Figure 6: Performance curves per query size for normal (un-encoded) DNA

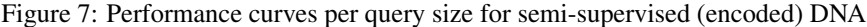

Figure 7: Performance curves per query size for semi-supervised (encoded) DNA

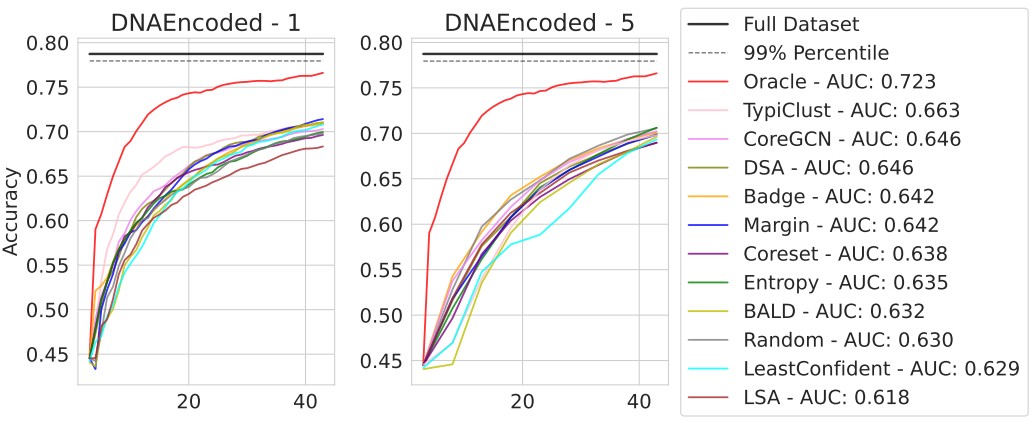

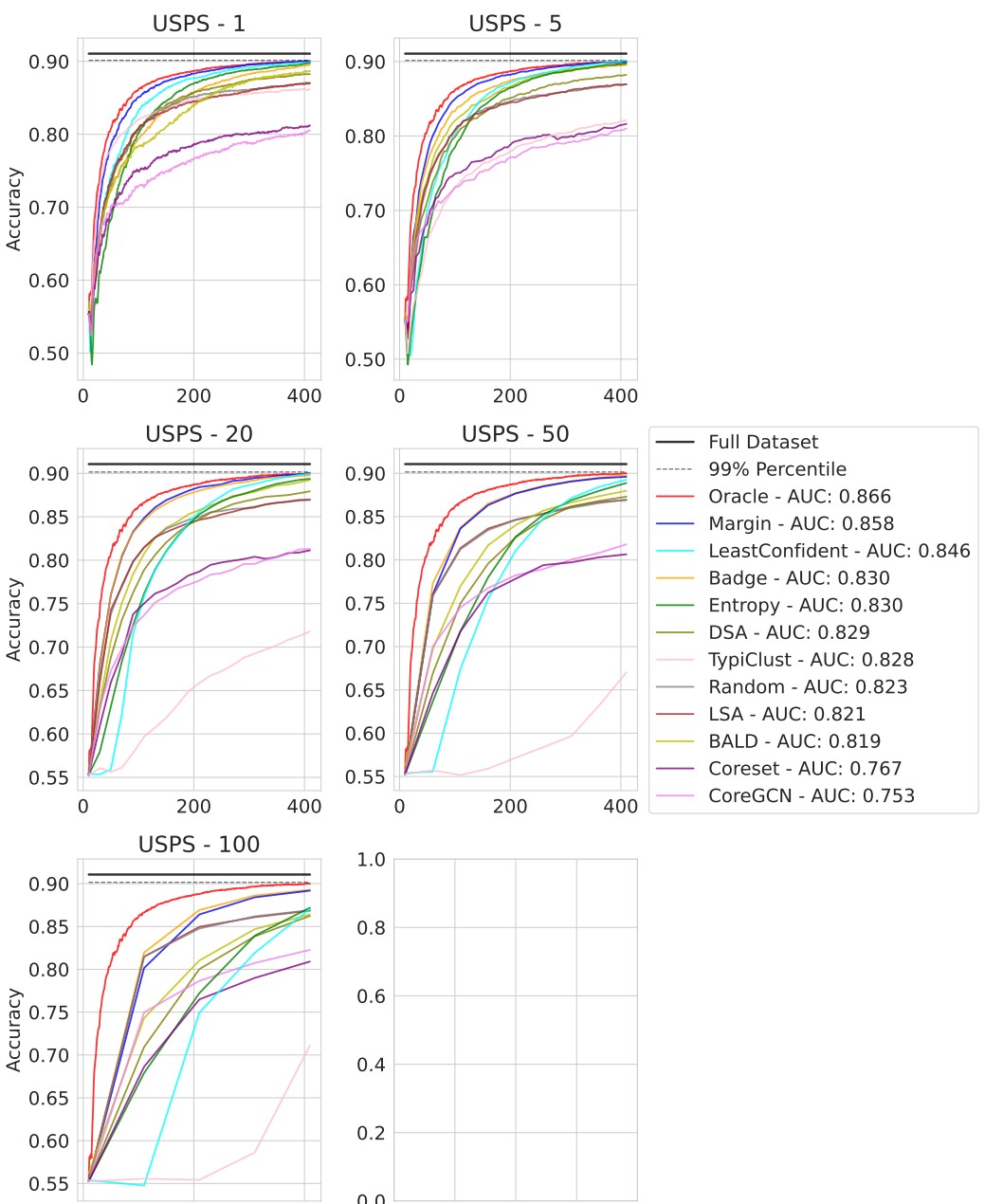

Figure 8: Performance curves per query size for normal (un-encoded) USPS

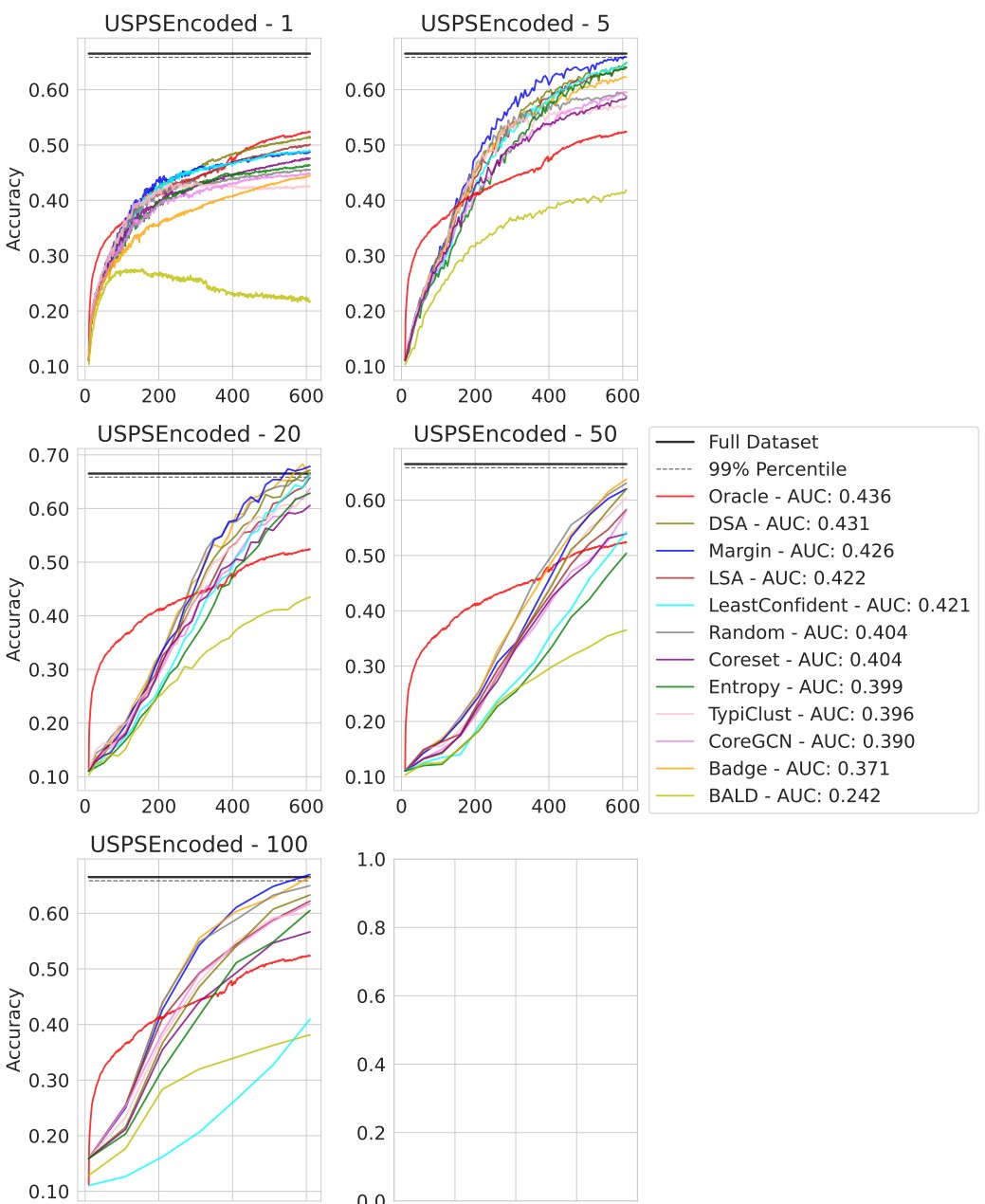

Figure 9: Performance curves per query size for semi-supervised (encoded) USPS

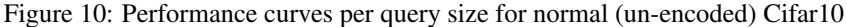

Figure 10: Performance curves per query size for normal (un-encoded) Cifar10

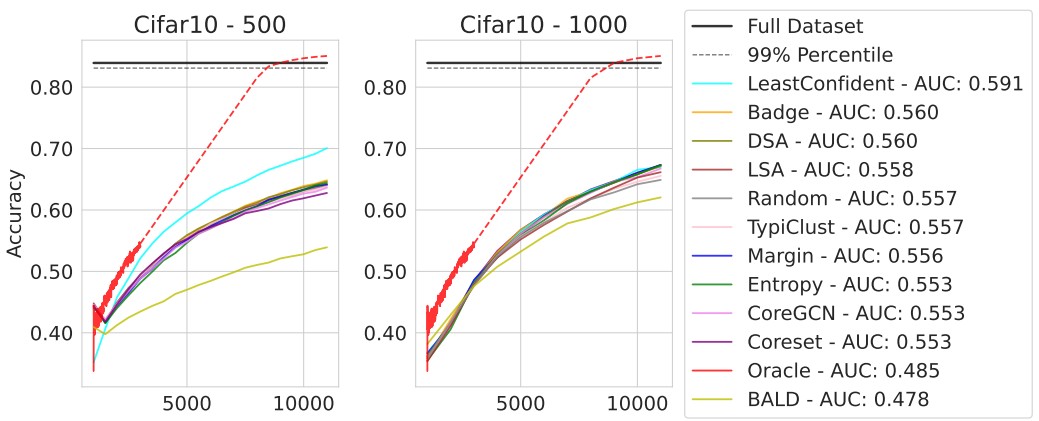

Figure 11: Performance curves per query size for semi-supervised (encoded) Cifar10

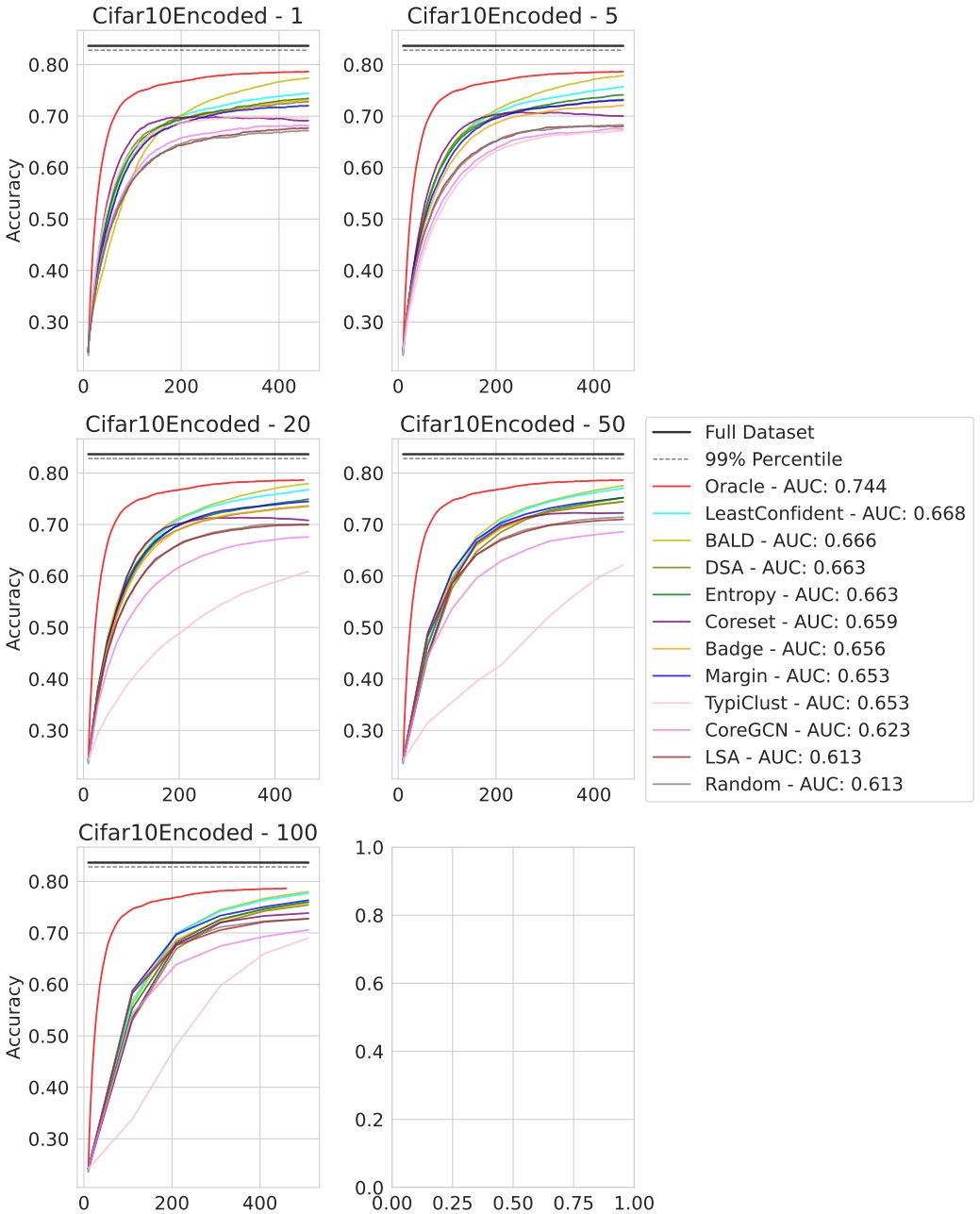

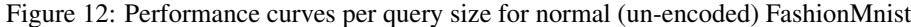

Figure 12: Performance curves per query size for normal (un-encoded) FashionMnist

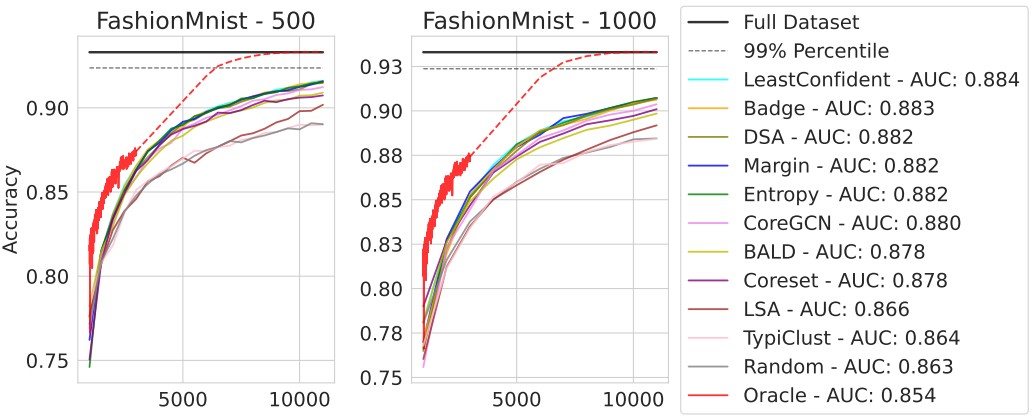

Figure 13: Performance curves per query size for semi-supervised (encoded) FashionMnist

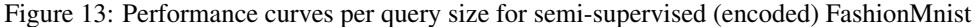

Figure 14: Performance curves per query size for normal (GloVe) TopV2

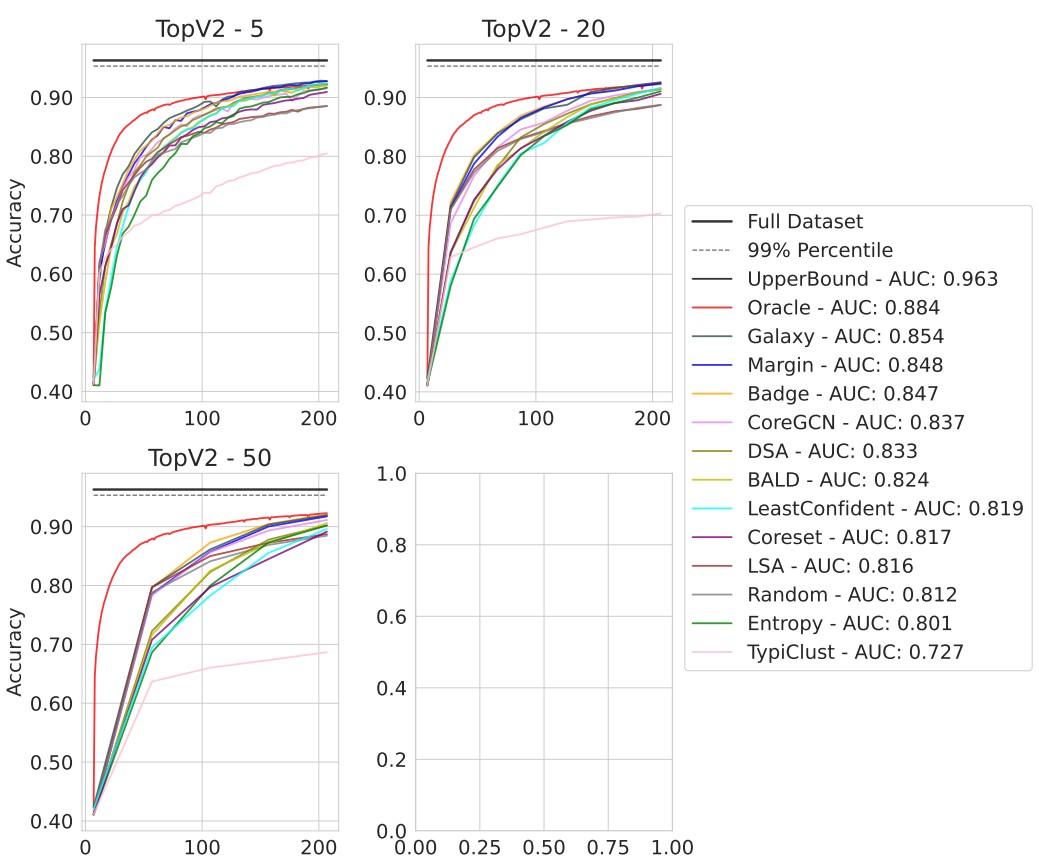

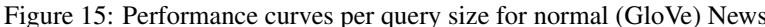

Figure 15: Performance curves per query size for normal (GloVe) News

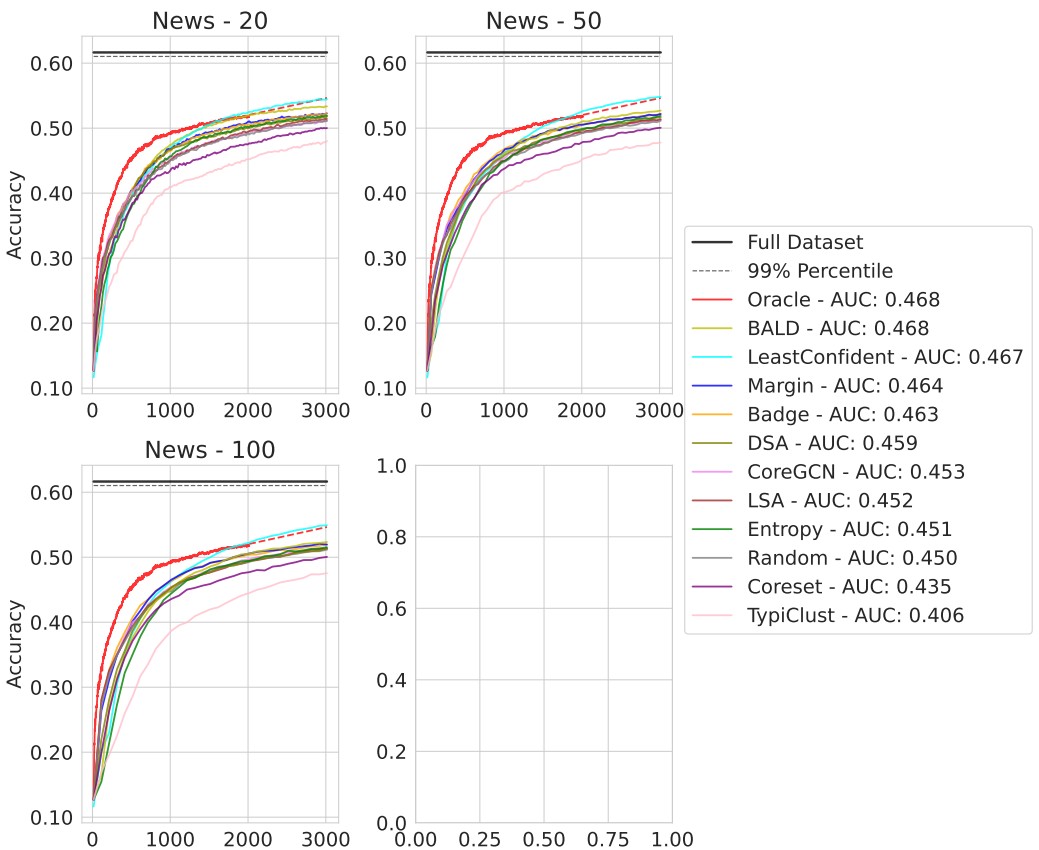

Figure 16: Performance curves per query size for normal (un-encoded) Honeypot

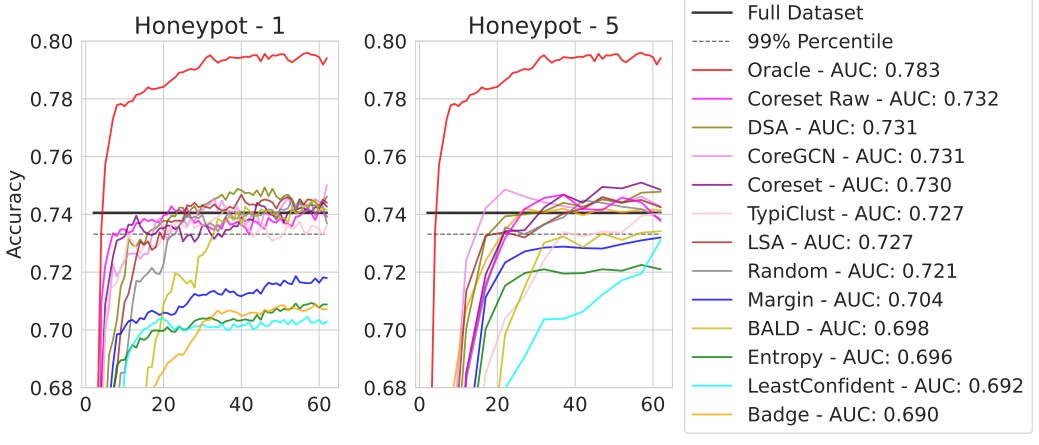

Figure 17: Performance curves per query size for normal (un-encoded) Diverging Sine

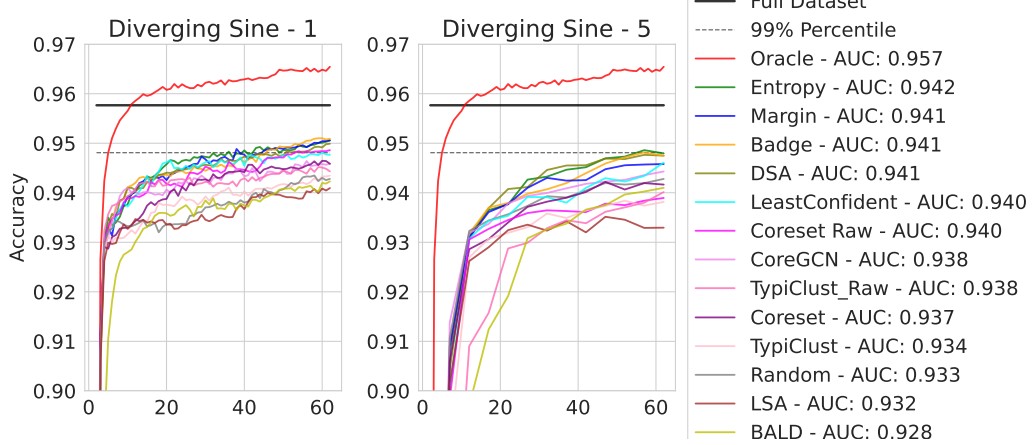

## G    Critical Difference Diagrams

We adapted the code for the CD diagrams from [8].
To compare each AL method, we consider each combination of dataset, query size and run is considered a separate "experiment", i.e. the results of `Dataset1-QuerySize1-run5` of an AL Method `x` is only compared to the results of `Dataset1-QuerySize1-run5` of AL method `y`.
Depending on the use-case, we build the following "experiments":

- Single dataset - single query size: Each AL method has 50 "experiments" in it's 50 repetitions

- Single dataset - all query sizes (Fig. 3): Each "experiment" is represented by a string `query_size_<qs>_run_<id>`

- Multiple dataset - all query sizes (Fig. 2): Each "experiment" is represented by a string `dataset_<dataset>_query_size_<qs>_run_<id>`

Due to the large number of restarts and the wide range of datasets and query sizes, we can provide very accurate significance tests.

## H    AL Pseudocode

---
**Algorithm 1** Active Learning Loop

---
**Require:** $\mathcal{L}, \mathcal{U}, \mathcal{D}_{\text{test}}, \text{Train}, \text{Seed}, \hat{y}$
**Require:** $\Omega$             ▷ AL Method
 1: $\mathcal{L}^{(1)} \leftarrow \text{Seed}(\mathcal{U})$        ▷ Create the initial labeled set
 2: $\mathcal{U}^{(1)} \leftarrow \mathcal{U}$
 3: **for** $i := 1 \dots B$ **do**
 4:      $\text{acc}^{(i)} \leftarrow \text{Train}(\mathcal{L}^{(i)})$
 5:      $a^{(i)} \leftarrow \Omega(\mathcal{U}^{(i)})$
 6:      $\mathcal{L}^{(i+1)} \leftarrow \mathcal{L}^{(i)} \cup \{(\mathcal{U}_a^{(i)}, A(\mathcal{U}_a^{(i)}))\}$
 7:      $\mathcal{U}^{(i+1)} \leftarrow \mathcal{U}^{(i)} \setminus \{\mathcal{U}_a^{(i)}\}$
 8: **return** $\frac{1}{B} \sum_{i=1}^{B} \text{acc}^{(i)}$

---

---
**Algorithm 2** Retrain
---
**Require:** $\mathcal{L}, \mathcal{D}_{\text{val}}, \mathcal{D}_{\text{test}}$
**Require:** $\hat{y}, e_{\max}$
1: $\text{loss}^* \leftarrow \infty$
2: **for** $i := 1 \ldots e^{\max}$ **do**
3:      $\hat{y}_{i+1} \leftarrow \hat{y}_i - \eta \nabla_{\hat{y}} \ell(\mathcal{L}, \hat{y})$
4:      $\text{loss}_i \leftarrow \ell(\mathcal{D}^{\text{val}}, \hat{y})$
5:      **if** $\text{loss}_i < \text{loss}^*$ **then**
6:          $\text{loss}^* \leftarrow \text{loss}_i$
7:      **else**
8:          Break
9: **return** $\text{Acc}(\mathcal{D}^{\text{test}}, \hat{y})$
---

---
**Algorithm 3** Acquire Oracle $\Omega$
---
**Require:** $\mathcal{U}, \mathcal{L}, A, \mathcal{D}_{\text{test}}, \tau, \hat{y}_\theta$
**Require:** Train, Margin, Acc
1: $\text{acc}^0 \leftarrow \text{acc}^* \leftarrow \text{Acc}(\mathcal{D}_{\text{test}}, \hat{y}_\theta)$
2: **for** $k := 1 \ldots \tau$ **do**
3:      $u_k = \text{unif}(\mathcal{U})$
4:      $\mathcal{L}' \leftarrow \mathcal{L}^{(i)} \cup \{(u_k, A(u_k))\}$
5:      $\hat{y}'_\theta \leftarrow \text{Train}(\mathcal{L}', \hat{y}_\theta)$
6:      $\text{acc}' \leftarrow \text{Acc}(\mathcal{D}_{\text{test}}, \hat{y}'_\theta)$
7:      **if** $\text{acc}' > \text{acc}^*$ **then**
8:          $\text{acc}^* \leftarrow \text{acc}'$
9:          $u^* \leftarrow u_k$
10: **if** $\text{acc}^0 = \text{acc}^*$ **then**
11:      $u^* \leftarrow \text{Margin}(\mathcal{U}, \hat{y}_\theta)$
     **return** $u^*$
---

Alg. 3 replaces the AL method $\Omega$ in the AL loop (Alg. H line 5).

# I   Oracle Curve Forecasting

Unfortunately, the iterative nature of our oracle means that the computational effort scales in the budget $O(\tau B)$. For datasets with large budgets, like Cifar10 and FashionMnist (both 10K), we were unable to compute the oracle set for the entire 10K iterations.

We compromised by (i) picking the two points with highest test accuracy, instead of only one and (ii) only computed until iteration 2000.

The rest of the curve was forecast using a simple 2-step algorithm, based on linear regression:

1. Fit a linear regression model on the second 50% of the existing oracle curve (to accurately capture the trend of the oracle, rather than the intercept) and forecast the oracle performance for the remaining budget.

2. Post-process the oracle forecast by letting it asymptotically approach the upper bound performance of the dataset.

$$o_i = \min \begin{cases} o_i \\ \phi(i) * o_i + (1 - \phi(i)) * \text{upper bound} \end{cases} \tag{2}$$
$$\phi(i) = e^{-i/0.5} \tag{3}$$

Figure 18: (left) Oracle forecast (dotted line) for FashionMnist with query size 500; (right) function $\phi$ that governs the approach towards the upper bound performance.

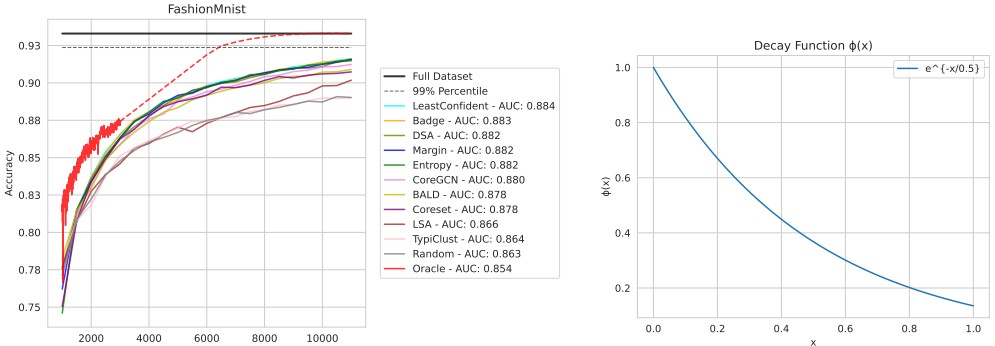

# J AL Methods

**Uncertainty Sampling** tries to find the sample that the classifier is most uncertain about by computing heuristics of the class probabilities. For our benchmark, we use entropy and margin (a.k.a. best-vs-second-best) sampling.

**BALD [15]** applies the query-by-committee strategy of model ensembles to a single model by interpreting the classifier's parameters as distributions and then sample multiple outputs from them via Monte-Carlo dropout.

**BADGE [1]** uses gradient embeddings of unlabeled points to select samples where the classifier is expected to change a lot. The higher the magnitude of the gradient the higher the expected improvement in model performance. BADGE employs a variant to the KMeans++ initialization technique to select batches of points. Even though [1] provided pseudocode for this procedure that selects the first point at random, all found implementations of BADGE select the first points instead by maximum gradient magnitude.

**Coreset [26]** employs K-Means clustering trying to cover the whole data distribution. Selects the unlabeled sample that is the furthest away from all cluster centers. Clustering is done in a semantically meaningful space by encoding the data with the current classifier $\hat{y}$. In this work, we use the greedy variant of Coreset.

**TypiClust [10]** relies on clustering similar to Coreset, but proposes a new measure called "Typicality" to select unlabeled samples. It selects points that are in the densest regions of clusters that do not contain labeled samples yet. Clustering is done in a semantically meaningful space by encoding the data with the current classifier $\hat{y}$. It has to be pointed out that TypiClust was designed for low-budget scenarios, but we think it is still worthwhile to test and compare this method with higher budgets.

**Core-GCN [4]** trains a Graph-Convolutional-Network (GCN) on embeddings of the unlabeled pool, obtained from the classifier (Similar to Coreset and TypiClust). This GCN model propagates uncertainty information through the graph and therefore enhances the nodes uncertainty quantification. Lastly, the node that displays the highest amount of uncertainty is selected for labeling.

**DSA/LSA [14]** use the metric of test adequacy to construct a set of points that is diverse, ranging from points that are close to points in $\mathcal{L}$ and points that are significantly different. DSA and LSA measure the diversity of points by distance in embedding space or likelihood estimation under the given classifier respectively.

**GALAXY [33]** construct a graph, where every point in $\mathcal{U}$ is a node and the edges are constructed from the current classifiers output. The algorithm then queries points from $\mathcal{U}$ based on edges in the graph, that are connected to two nodes with different predicted labels.

**Excluded Methods**

**Learning Loss for AL [29]** Introduces an updated training of the classification model with an auxiliary loss and therefore cannot be compared fairly against classification models without this boosted training regime.

**Reinforcement Learning Methods**

We postpone the study of learned AL methods to future versions of this benchmark, as reinforcement learning is infamous for being extremely time consuming and itself hard to reproduce .

# K AUCs by Query Size

All tables are sorted according to the main result in Table 3.

Table 8: AUC values for each dataset that supports query size 1.

| | Wins | Splice | SpliceEnc | DNA | DNAEnc | USPS | USPSEnc | Cifar10Enc | FMnistEnc | TopV2 | Diverging Sine | ThreeClust |
|---|---|---|---|---|---|---|---|---|---|---|---|---|
| Oracle | | 0.803±0.012 | 0.678±0.021 | 0.825±0.009 | 0.723±0.015 | 0.866±0.004 | 0.436±0.057 | 0.749±0.009 | 0.755±0.005 | 0.884±0.006 | 0.957±0.009 | 0.783±0.03 |
| Margin | 0 | 0.769±0.021 | 0.678±0.032 | 0.806±0.013 | 0.642±0.047 | 0.858±0.006 | 0.426±0.038 | 0.653±0.013 | 0.68±0.012 | 0.861±0.009 | 0.941±0.018 | 0.704±0.074 |
| Galaxy | 2 | 0.777±0.02 | 0.68±0.032 | 0.803±0.009 | 0.64±0.043 | **0.859±0.005** | 0.42±0.042 | 0.657±0.013 | 0.68±0.01 | **0.863±0.01** | 0.941±0.016 | 0.687±0.096 |
| Badge | 0 | 0.767±0.02 | 0.661±0.026 | 0.78±0.014 | 0.642±0.046 | 0.83±0.008 | 0.371±0.035 | 0.656±0.013 | 0.68±0.009 | 0.826±0.024 | 0.941±0.017 | 0.69±0.083 |
| LeastConfident | 3 | **0.779±0.019** | 0.68±0.032 | 0.809±0.01 | 0.629±0.05 | 0.846±0.009 | 0.421±0.039 | **0.668±0.014** | **0.685±0.009** | 0.843±0.013 | 0.94±0.016 | 0.692±0.094 |
| DSA | 4 | 0.766±0.021 | **0.691±0.022** | 0.803±0.01 | **0.646±0.032** | 0.829±0.01 | **0.431±0.05** | 0.663±0.014 | 0.679±0.01 | 0.844±0.017 | 0.941±0.014 | **0.731±0.032** |
| BALD | 0 | 0.78±0.014 | 0.649±0.04 | 0.784±0.01 | 0.632±0.042 | 0.819±0.01 | 0.242±0.046 | 0.666±0.014 | 0.644±0.018 | 0.815±0.024 | 0.928±0.014 | 0.698±0.043 |
| CoreGCN | 4 | 0.765±0.021 | 0.686±0.023 | 0.804±0.012 | **0.646±0.03** | 0.753±0.016 | 0.39±0.044 | 0.623±0.018 | 0.647±0.012 | 0.85±0.01 | 0.938±0.014 | **0.731±0.028** |
| Entropy | 2 | 0.768±0.022 | 0.678±0.035 | **0.812±0.013** | 0.635±0.045 | 0.83±0.011 | 0.399±0.035 | 0.663±0.013 | 0.681±0.011 | 0.815±0.021 | **0.942±0.017** | 0.696±0.083 |
| LSA | 0 | 0.772±0.016 | 0.68±0.026 | 0.787±0.012 | 0.618±0.036 | 0.821±0.009 | 0.422±0.037 | 0.613±0.014 | 0.642±0.012 | 0.816±0.013 | 0.932±0.016 | 0.727±0.033 |
| Random | 0 | 0.76±0.016 | 0.674±0.027 | 0.774±0.013 | 0.63±0.035 | 0.823±0.009 | 0.404±0.036 | 0.613±0.014 | 0.639±0.013 | 0.815±0.012 | 0.933±0.017 | 0.721±0.036 |
| Coreset | 0 | 0.772±0.016 | 0.69±0.017 | 0.79±0.012 | 0.638±0.041 | 0.767±0.016 | 0.404±0.046 | 0.659±0.011 | 0.684±0.009 | 0.826±0.022 | 0.937±0.014 | 0.73±0.031 |
| TypiClust | 0 | 0.762±0.016 | 0.685±0.025 | 0.778±0.01 | 0.663±0.028 | 0.828±0.007 | 0.396±0.046 | 0.653±0.013 | 0.649±0.007 | 0.831±0.011 | 0.934±0.018 | 0.727±0.033 |

Table 9: AUC values for each dataset that supports query size 5.

| | Wins | Splice | SpliceEnc | DNA | DNAEnc | USPS | USPSEnc | Cifar10Enc | FMnistEnc | TopV2 | DivergingSin | ThreeClust |
|---|---|---|---|---|---|---|---|---|---|---|---|---|
| Oracle | | 0.803±0.012 | 0.678±0.021 | 0.825±0.009 | 0.723±0.015 | 0.866±0.004 | 0.436±0.057 | 0.749±0.009 | 0.755±0.005 | 0.884±0.006 | 0.957±0.009 | 0.783±0.03 |
| Margin | 2 | 0.765±0.021 | 0.662±0.032 | 0.794±0.011 | 0.611±0.05 | **0.855±0.006** | **0.508±0.02** | 0.656±0.014 | 0.678±0.009 | 0.848±0.013 | 0.923±0.019 | 0.697±0.055 |
| Galaxy | 2 | 0.758±0.025 | 0.647±0.042 | 0.799±0.011 | 0.611±0.043 | **0.855±0.006** | 0.506±0.023 | 0.661±0.012 | 0.677±0.01 | **0.854±0.01** | 0.922±0.02 | 0.674±0.077 |
| Badge | 2 | 0.768±0.014 | 0.646±0.035 | 0.785±0.011 | **0.624±0.036** | 0.846±0.007 | 0.48±0.021 | 0.647±0.012 | 0.67±0.009 | 0.847±0.01 | **0.924±0.019** | 0.72±0.036 |
| LeastConfident | 1 | 0.763±0.023 | 0.643±0.034 | 0.798±0.013 | 0.585±0.065 | 0.831±0.014 | 0.478±0.028 | 0.67±0.01 | 0.681±0.009 | 0.819±0.023 | 0.921±0.019 | 0.675±0.072 |
| DSA | 1 | 0.765±0.023 | 0.653±0.029 | 0.793±0.009 | 0.613±0.034 | 0.822±0.01 | 0.489±0.024 | 0.661±0.013 | 0.662±0.012 | 0.833±0.02 | **0.924±0.018** | 0.718±0.033 |
| BALD | 3 | 0.775±0.018 | 0.641±0.034 | **0.801±0.013** | 0.592±0.054 | 0.84±0.008 | 0.332±0.054 | **0.681±0.009** | **0.681±0.011** | 0.824±0.023 | 0.893±0.035 | 0.673±0.041 |
| CoreGCN | 1 | 0.759±0.018 | 0.662±0.027 | 0.79±0.011 | 0.62±0.03 | 0.755±0.011 | 0.45±0.03 | 0.604±0.016 | 0.609±0.013 | 0.837±0.014 | 0.922±0.018 | **0.723±0.034** |
| Entropy | 1 | 0.765±0.022 | 0.66±0.03 | 0.798±0.011 | 0.611±0.054 | 0.823±0.013 | 0.464±0.024 | 0.663±0.013 | 0.672±0.011 | 0.801±0.025 | **0.924±0.02** | 0.689±0.066 |
| LSA | 1 | **0.769±0.016** | 0.654±0.032 | 0.781±0.013 | 0.61±0.041 | 0.82±0.009 | 0.484±0.022 | 0.617±0.012 | 0.641±0.011 | 0.816±0.012 | 0.915±0.018 | 0.718±0.038 |
| Random | 0 | 0.758±0.015 | 0.655±0.026 | 0.771±0.013 | 0.623±0.031 | 0.82±0.009 | 0.476±0.024 | 0.616±0.016 | 0.637±0.012 | 0.812±0.014 | 0.921±0.018 | 0.713±0.034 |
| Coreset | 1 | 0.765±0.017 | **0.663±0.023** | 0.784±0.014 | 0.603±0.034 | 0.765±0.015 | 0.449±0.022 | 0.657±0.009 | 0.674±0.009 | 0.817±0.017 | 0.92±0.017 | 0.713±0.035 |
| TypiClust | 0 | 0.759±0.014 | 0.641±0.028 | 0.775±0.01 | 0.603±0.04 | 0.757±0.02 | 0.465±0.027 | 0.596±0.014 | 0.567±0.012 | 0.727±0.026 | 0.916±0.02 | 0.693±0.045 |

Table 10: AUC values for each dataset that supports query size 20.

| | Wins | Splice | SpliceEnc | DNA | USPS | USPSEnc | Cifar10Enc | FMnistEnc | TopV2 | News |
|---|---|---|---|---|---|---|---|---|---|---|
| Oracle | | 0.803±0.012 | 0.678±0.021 | 0.825±0.009 | 0.866±0.004 | 0.436±0.057 | 0.749±0.009 | 0.755±0.005 | 0.884±0.006 | 0.49±0.003 |
| Margin | 1 | 0.759±0.027 | 0.618±0.04 | **0.779±0.013** | 0.847±0.008 | 0.439±0.027 | 0.656±0.01 | 0.67±0.011 | 0.823±0.014 | 0.464±0.007 |
| Galaxy | 3 | 0.751±0.023 | 0.599±0.05 | 0.761±0.023 | **0.848±0.009** | **0.444±0.022** | 0.662±0.012 | 0.669±0.011 | 0.826±0.018 | **0.483±0.007** |
| Badge | 2 | 0.767±0.013 | **0.619±0.033** | 0.776±0.013 | 0.845±0.006 | 0.44±0.019 | 0.647±0.013 | 0.665±0.007 | **0.827±0.016** | 0.463±0.007 |
| LeastConfident | 1 | 0.751±0.02 | 0.597±0.05 | 0.748±0.025 | 0.798±0.027 | 0.391±0.024 | **0.665±0.013** | 0.669±0.011 | 0.775±0.035 | 0.467±0.008 |
| DSA | 0 | 0.759±0.02 | 0.599±0.034 | 0.769±0.013 | 0.809±0.012 | 0.421±0.023 | 0.647±0.014 | 0.63±0.013 | 0.793±0.026 | 0.459±0.01 |
| BALD | 1 | **0.768±0.022** | 0.57±0.037 | 0.784±0.015 | 0.822±0.009 | 0.298±0.039 | 0.675±0.008 | 0.673±0.01 | 0.789±0.024 | 0.468±0.009 |
| CoreGCN | 0 | 0.759±0.018 | 0.612±0.039 | 0.774±0.012 | 0.754±0.016 | 0.397±0.026 | 0.587±0.015 | 0.583±0.015 | 0.807±0.018 | 0.453±0.006 |
| Entropy | 0 | 0.759±0.027 | 0.618±0.038 | 0.773±0.013 | 0.803±0.019 | 0.372±0.022 | 0.656±0.011 | 0.65±0.012 | 0.773±0.031 | 0.451±0.007 |
| LSA | 0 | 0.761±0.014 | 0.611±0.039 | 0.768±0.015 | 0.816±0.009 | 0.411±0.022 | 0.621±0.01 | 0.635±0.011 | 0.796±0.016 | 0.452±0.007 |
| Random | 0 | 0.755±0.014 | 0.612±0.039 | 0.763±0.012 | 0.818±0.009 | 0.439±0.019 | 0.622±0.013 | 0.633±0.012 | 0.795±0.016 | 0.45±0.006 |
| Coreset | 0 | 0.759±0.016 | 0.601±0.034 | 0.764±0.015 | 0.757±0.015 | 0.39±0.029 | 0.647±0.009 | 0.651±0.011 | 0.784±0.026 | 0.435±0.012 |
| TypiClust | 0 | 0.751±0.012 | 0.551±0.036 | 0.76±0.016 | 0.643±0.026 | 0.411±0.024 | 0.488±0.02 | 0.449±0.017 | 0.652±0.035 | 0.406±0.011 |

Table 11: AUC values for each dataset that supports query size 50.

| | Wins | Splice | DNA | USPS | USPSEnc | Cifar10Enc | FMnistEnc | TopV2 | News |
|---|---|---|---|---|---|---|---|---|---|
| Oracle | | 0.803±0.012 | 0.825±0.009 | 0.866±0.004 | 0.436±0.057 | 0.749±0.009 | 0.755±0.005 | 0.884±0.006 | 0.49±0.003 |
| Margin | 0 | 0.747±0.023 | 0.751±0.019 | 0.828±0.009 | 0.363±0.031 | 0.64±0.013 | 0.653±0.01 | 0.774±0.029 | 0.46±0.006 |
| Galaxy | 4 | 0.733±0.027 | 0.702±0.037 | 0.828±0.01 | 0.363±0.029 | **0.647±0.009** | **0.654±0.008** | **0.781±0.029** | **0.482±0.006** |
| Badge | 3 | 0.748±0.017 | 0.754±0.018 | **0.831±0.008** | **0.376±0.028** | 0.632±0.013 | 0.649±0.011 | **0.781±0.026** | 0.462±0.007 |
| LeastConfident | 0 | 0.731±0.025 | 0.688±0.041 | 0.761±0.037 | 0.291±0.03 | 0.644±0.013 | 0.65±0.011 | 0.73±0.049 | 0.462±0.009 |
| DSA | 0 | 0.748±0.021 | 0.738±0.018 | 0.783±0.016 | 0.346±0.027 | 0.624±0.014 | 0.588±0.016 | 0.748±0.041 | 0.45±0.011 |
| BALD | 2 | **0.76±0.017** | **0.756±0.018** | 0.796±0.015 | 0.241±0.026 | 0.65±0.009 | 0.645±0.01 | 0.746±0.038 | 0.455±0.007 |
| CoreGCN | 0 | 0.755±0.016 | 0.745±0.018 | 0.752±0.019 | 0.328±0.027 | 0.581±0.015 | 0.568±0.018 | 0.771±0.025 | 0.453±0.007 |
| Entropy | 0 | 0.747±0.024 | 0.748±0.018 | 0.778±0.024 | 0.275±0.026 | 0.633±0.011 | 0.625±0.012 | 0.734±0.036 | 0.442±0.007 |
| LSA | 0 | 0.754±0.013 | 0.749±0.019 | 0.807±0.01 | 0.341±0.023 | 0.613±0.012 | 0.625±0.01 | 0.763±0.025 | 0.45±0.006 |
| Random | 0 | 0.746±0.012 | 0.745±0.015 | 0.806±0.008 | 0.379±0.028 | 0.615±0.014 | 0.621±0.01 | 0.759±0.026 | 0.448±0.006 |
| Coreset | 0 | 0.751±0.016 | 0.733±0.019 | 0.74±0.017 | 0.325±0.034 | 0.624±0.012 | 0.608±0.013 | 0.731±0.045 | 0.432±0.012 |
| TypiClust | 0 | 0.749±0.016 | 0.736±0.016 | 0.586±0.038 | 0.348±0.027 | 0.451±0.024 | 0.375±0.022 | 0.614±0.046 | 0.397±0.012 |

Table 12: AUC values for each dataset that supports query size 100.

| | Wins | Splice | DNA | USPS | USPSEnc | Cifar10Enc | FMnistEnc | News |
|---|---|---|---|---|---|---|---|---|
| Oracle | | 0.803±0.012 | 0.825±0.009 | 0.866±0.004 | 0.436±0.057 | 0.749±0.009 | 0.755±0.005 | 0.49±0.003 |
| Margin | 1 | 0.733±0.024 | 0.711±0.027 | 0.799±0.013 | **0.473±0.026** | 0.629±0.012 | 0.628±0.009 | 0.455±0.006 |
| Galaxy | 4 | 0.715±0.032 | 0.658±0.047 | **0.804±0.015** | 0.307±0.04 | **0.638±0.01** | **0.632±0.009** | **0.478±0.007** |
| Badge | 1 | 0.743±0.014 | 0.714±0.032 | **0.804±0.013** | 0.472±0.029 | 0.623±0.01 | 0.621±0.01 | 0.456±0.006 |
| LeastConfident | 0 | 0.715±0.033 | 0.639±0.05 | 0.708±0.034 | 0.23±0.034 | 0.631±0.013 | 0.62±0.012 | 0.457±0.008 |
| DSA | 0 | 0.729±0.021 | 0.697±0.031 | 0.753±0.021 | 0.427±0.028 | 0.609±0.013 | 0.546±0.017 | 0.442±0.01 |
| BALD | 2 | **0.744±0.015** | **0.718±0.024** | 0.765±0.021 | 0.285±0.046 | 0.632±0.009 | 0.609±0.01 | 0.444±0.007 |
| CoreGCN | 0 | 0.742±0.015 | 0.713±0.025 | 0.744±0.019 | 0.433±0.032 | 0.583±0.013 | 0.554±0.015 | 0.448±0.007 |
| Entropy | 0 | 0.733±0.023 | 0.713±0.031 | 0.743±0.026 | 0.395±0.037 | 0.618±0.012 | 0.59±0.012 | 0.432±0.007 |
| LSA | 0 | 0.738±0.017 | 0.716±0.027 | 0.789±0.011 | 0.439±0.03 | 0.609±0.013 | 0.608±0.01 | 0.447±0.006 |
| Random | 0 | 0.733±0.013 | 0.713±0.023 | 0.789±0.012 | 0.468±0.024 | 0.611±0.01 | 0.606±0.01 | 0.446±0.005 |
| Coreset | 0 | 0.735±0.019 | 0.698±0.026 | 0.721±0.021 | 0.396±0.024 | 0.608±0.012 | 0.562±0.016 | 0.426±0.012 |
| TypiClust | 0 | 0.733±0.016 | 0.704±0.025 | 0.592±0.042 | 0.427±0.027 | 0.501±0.02 | 0.338±0.02 | 0.383±0.012 |

Table 13: AUC values for each dataset that supports query size 500.

| | Cifar10 | FashionMnist | News |
|---|---|---|---|
| Oracle | 0.689±0.001 | 0.905±0.001 | 0.49±0.003 |
| Margin | 0.556±0.008 | 0.882±0.004 | 0.441±0.011 |
| Galaxy | **0.591±0.011** | 0.881±0.008 | 0.447±0.009 |
| Badge | 0.56±0.008 | 0.883±0.005 | **0.451±0.008** |
| LeastConfident | **0.591±0.01** | **0.884±0.005** | 0.425±0.013 |
| DSA | 0.56±0.009 | 0.882±0.004 | 0.432±0.014 |
| BALD | 0.478±0.014 | 0.878±0.003 | 0.421±0.012 |
| CoreGCN | 0.553±0.01 | 0.88±0.007 | 0.437±0.012 |
| Entropy | 0.553±0.009 | 0.882±0.006 | 0.419±0.015 |
| LSA | 0.558±0.01 | 0.866±0.005 | 0.437±0.009 |
| Random | 0.557±0.01 | 0.863±0.005 | 0.437±0.008 |
| Coreset | 0.553±0.007 | 0.878±0.006 | 0.439±0.009 |
| TypiClust | 0.557±0.009 | 0.864±0.004 | 0.43±0.01 |

Table 14: AUC values for each dataset that supports query size 1000.

| | Cifar10 | FashionMnist |
|---|---|---|
| Oracle | 0.689±0.001 | 0.905±0.001 |
| Margin | 0.56±0.011 | 0.872±0.007 |
| Galaxy | 0.56±0.011 | 0.87±0.007 |
| Badge | **0.562±0.013** | 0.871±0.007 |
| LeastConfident | 0.561±0.012 | **0.873±0.006** |
| DSA | 0.56±0.011 | 0.87±0.008 |
| BALD | 0.535±0.011 | 0.866±0.003 |
| CoreGCN | 0.557±0.011 | 0.867±0.012 |
| Entropy | 0.557±0.014 | 0.871±0.009 |
| LSA | 0.551±0.012 | 0.854±0.009 |
| Random | 0.55±0.01 | 0.855±0.006 |
| Coreset | 0.562±0.012 | 0.869±0.004 |
| TypiClust | 0.552±0.011 | 0.854±0.009 |

## L  Analysis of Results for the Semi-Supervised Domain

Even though the results for the aggregated semi-supervised domain appear in line with our overall ranking of methods, we observe stark differences for the sub-domains of semi-supervised image and semi-supervised tabular.

While semi-supervised images seem to mostly mirror the results from the normal image domain (with the exception of BALD), semi-supervised tabular data display highly irregular behavior, placing random sampling as second-best method behind the margin sampling. Our oracle method even falls behind other methods. Almost all methods a bunched into one region in the CD diagram with many non-significance bars indicating few, if any, significant differences between the methods.

Both the reasons, for the sub-random performance of most methods, and the bad performance of our oracle are currently unknown and require further research.

Figure 19: Results for the semi-supervised domain, aggregated over all data types (top) and separately for images and tabular (bottom)

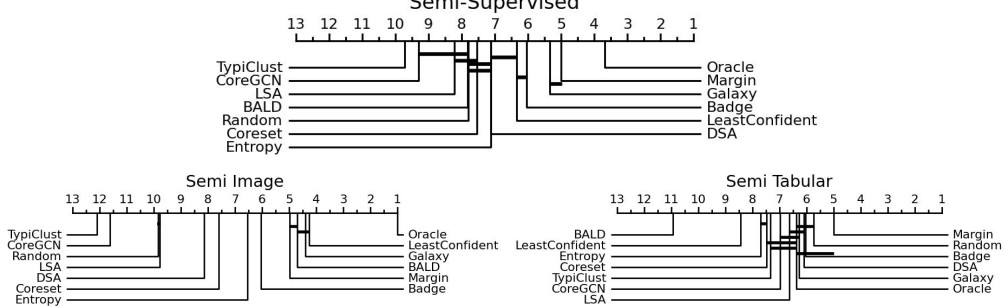

## M    Hyperparameters per AL Method

Table 15: Selected hyperparameters for all tested AL methods. Last column indicates the source of our implementation.

| Method | Sample Size | Other | Source |
|---|---|---|---|
| BADGE | 5000 | | Based on [1, 17] |
| BALD | 5000 | Dropout Trials: 5 | Based on [5] |
| Coreset | 5000 | | Own |
| TypiClust | 5000 | Min Cluster Size: 5 Max # Clusters: 500 | Based on [10] |
| Margin | 5000 | | Own |
| Entropy | 5000 | | Own |

## N    Amount of Computational Resources Invested

For our results we computed a total of 24200 runs (without Oracle runs) over a span of  4 months. We used our computational cluster consisting of 30-40 GPUs.

Number of runs per dataset and query size : 11 alg. * 50 runs = 550

Runs per dataset:
Cifar10 x2 = 1100
Cfr10Enc x4 = 2200
DivSin x2 = 1100
DNA x4 = 2200
DNAEnc x2 = 1100
FMnist x2 = 1100
FMnistEnc x4 = 2200
News x3 = 1650
Splice x4 = 2200
SpliceEnc x3 = 1650
Honeypot x2 = 1100
TopV2 x4 = 2200
USPS x4 = 2200
USPSEnc x4 = 2200
= 24200 runs

## O Limitations and Future Work

Even though our benchmark includes a wide range of data domains, the number of datasets per domain is still limited. It remains untested if our selected datasets are indeed a good representation of their domain, or if additional datasets would skew the results of Fig. 2.

Additionally, since we began working on this benchmark a few new AL Methods have been published. We consciously focused on only those methods for which good results have been reported by multiple sources, consequently omitting the newest methods.

Most obviously, our future work involves the implementation of more datasets per domain and the newest AL methods.

The choice of ranks for the main result table, like any other choice, has an impact in the interpretation of the results. E.g. comparing the rank of BALD in Table 3 (rank 6.6) with the amount of wins it is able to obtain in the AUC-based tables in Appendix K, suggests that an evaluation purely based on mean AUC values would count BALD to the best methods.

We advocate for using ranks in AL evaluations for two reasons: (i) they are more robust to outlier performances in single runs and (ii) they highlight wether an method is able to consistently outperform another method, even if the difference in mean AUC is very small.

Nonetheless, we think that the topic of truly fair evaluations for AL needs further research.

Lastly, it remains untested if the differences between domains that we observe, are truly caused by the differences in data, but could also be influenced by the type of model that is common to those domains.

An important part of our future work is therefore to test different model archetypes per domain.

