# OpenReview forum: "A Cross-Domain Benchmark for Active Learning"
_NeurIPS.cc/2024/Datasets_and_Benchmarks_Track — NeurIPS 2024 Track Datasets and Benchmarks Poster_

### Official Review · Reviewer_c7pZ · 2024-07-19

**Rating:** 7
**Confidence:** 5
**Correctness:** Yes, see reviews for comment around m…
**Clarity:** Yes, very clear.

**Review:**

I think the unification of multiple domains in an active learning benchmark is a great resource for researchers to test out their algorithms. The paper can be further improved as follows:
1. Include more related benchmarks. For example [1] studies extensively the behavior of active learning in tabular datasets. LabelBench [2] has also proposed using embedded features and semi-supervised learning as part of their benchmark. While the authors in this paper conjecture "a well-performing method in our benchmark will also generalize well to larger datasets and classifiers", LabelBench has already demonstrated this to be true. In addition, margin sampling performing well on tabular datasets is also found in [1].
2. The proposed metric of average ranking may not be a convenient/intuitive metric for benchmarks. Specifically, whenever a new algorithm is introduced, the scores of every algorithm will change. Moreover, practitioners are generally interested in either accuracy or label-efficiency (number of labels needed to reach a certain accuracy). The adopted metric does not capture any of these quantities directly. In some cases, entropy, badge and margin may all perform very similarly in accuracy/label-efficiency but remains stable in their relative ranking. The metric in this paper would significantly penalize the algorithms that perform relatively worse, which does not give an accurate quantification in the actual performance of the algorithms themselves.
3. An interesting finding in LabelBench is imbalanced active learning algorithms like GALAXY performs significantly better on imbalanced datasets. Since most of the language and tabular datasets are imbalanced, it would be interesting to see how such algorithms perform. The conclusion that "entropy sampling" or "margin sampling" is the best may not be entirely accurate.

[1] Bahri, D., Jiang, H., Schuster, T., & Rostamizadeh, A. (2022). Is margin all you need? An extensive empirical study of active learning on tabular data. arXiv preprint arXiv:2210.03822.
[2] Zhang, J., Chen, Y., Canal, G., Das, A. M., Bhatt, G., Mussmann, S., ... & Nowak, R. D. (2024). LabelBench: A Comprehensive Framework for Benchmarking Adaptive Label-Efficient Learning. Journal of Data-centric Machine Learning Research.

**Strengths:**

See review

**Additional Feedback:**

N/A

**Documentation:**

The codebase appears to need more documentation around how to launch an experiment and in code files in general. The codebase could also benefit from having several example files.

**Limitations:**

Yes

**Opportunities For Improvement:**

See review

**Relation To Prior Work:**

See bullet 1 in reviews

**Summary And Contributions:**

The paper presents a new benchmark for active learning across three domains of datasets including vision, language and tabular. The authors also propose a subset selection algorithm to serve as upper bound of active learning performance. The results suggest that
1) averaging over only three trials in common experimental practice may significantly hinder the accuracy in relative performance between algorithms
2) the best active learning strategy is not consistent across domains, suggesting one should potentially use different algorithms for different domains.

---

> ### Author Rebuttal · Authors · 2024-08-14
>
> We thank the reviewer for their detailed review and would like to address the individual points:
> 1. Include more related benchmarks. For example [1] [...] In addition, margin sampling performing well on tabular datasets is also found in [1].
> - [1] and [2] will be integrated into Table 1 and our related work section.
> Based on the comment of another reviewer we aim to extend our discussion of our results in the appendix, which will also include a more in-depth comparison of our results with other benchmarks like [1].
> 2. The proposed metric of average ranking may not be a convenient/intuitive metric for benchmarks. Specifically, whenever a new algorithm is introduced, the scores of every algorithm will change. [...]
> - We fully agree with the reviewer that accuracy scores need to be reported for any classification problem.
> We have all accuracy scores in Appendix K, including standard deviations and the amount of wins per algorithm.
> We would like to point out that our benchmark aggregates results from different datasets, which we cannot be done by simply averaging the accuracies to display them in a table, etc.
> For a fair comparison, we rely on the paired-t test, which the Critical-Difference-Diagrams conveniently provide.
> Lastly, we were unable to include a second results table besides Table 3 within the page limit and therefore located our accuracy values in the Appendix.
> 3. An interesting finding in LabelBench is imbalanced active learning algorithms like GALAXY performs significantly better on imbalanced datasets. [...]
> - We omitted information about class imbalance in our datasets, as we did not focus on that aspect.
> However, both our text and 2 out of 3 tabular datasets are imbalanced by nature. Implementing GALAXY and discussing results along this dimension as well is a great suggestion.
> Since testing GALAXY on our entire benchmark is a significant computational effort, we can only provide experiments on the TopV2 dataset for this rebuttal.
> However, GALAXY is indeed the top-performing algorithm on the imbalanced TopV2 dataset (see uploaded PDF for results).
> We thank the reviewer for this valuable addition to the benchmark and will fully incorporate GALAXY for the camera-ready version.

---

> > ### Comment · Reviewer_c7pZ · 2024-08-26
> >
> > Thank you for your rebuttal. I have raised my score to a 7.

---

> ### Author Response · Authors · 2024-08-26
> **Thank you for your feedback**
>
> We again thank the reviewer for the valuable feedback and hope that we have addressed all concerns. As the author-discussion deadline approaches, please let us know if further clarification is needed.
> Best regards,
> The authors"

---

### Official Review · Reviewer_x7yL · 2024-07-24
**A comprehensive benchmark**

**Rating:** 6
**Confidence:** 5

**Review:**

Pros:
1. This work covers multiple domains and includes both real-life and synthetic datasets. Including synthetic datasets like Honeypot and Diverging Sine to test specific shortcomings of AL methods is interesting.
2. The paper gives very detailed explanations of the importance of multiple runs and cross-domain evaluations.

Cons:
1. Although the author states that the proposed benchmark is the first one that contains multiple domains, however, in each domain, many related empirical studies focus on specific domain, e.g., [r1], [r2], [r3], which weakens the contribution of this paper.
2. The insights (e.g., the importance of multiple runs) build on existing knowledge in the active learning field.
3. This paper takes too much space on the greedy oracle algorithm and seeding strategies, which may be complex for some readers to fully get the meaning.

[r1] Bahri, Dara, et al. "Is margin all you need? An extensive empirical study of active learning on tabular data." arXiv preprint arXiv:2210.03822 (2022).

[r2] Zhang, Zhisong, Emma Strubell, and Eduard Hovy. "A survey of active learning for natural language processing." arXiv preprint arXiv:2210.10109 (2022).

[r3] Schröder, Christopher, and Andreas Niekler. "A survey of active learning for text classification using deep neural networks." arXiv preprint arXiv:2008.07267 (2020).

**Strengths:**

This paper demonstrates the importance of multiple repetitions in obtaining meaningful conclusions, this is valuable, as it can help set a higher standard for the robustness of experimental results in AL.

**Additional Feedback:**

No need to show empty figures, e.g., Figure 4 in the appendix.

**Clarity:**

This paper has some issues in arranging the contents in the main manuscript, they put many valuable information in the appendix.

**Correctness:**

The evaluation methods and experiment design were appropriate and performed correctly.

**Documentation:**

Thre are sufficient details to support reproducibility.

**Limitations:**

The author states that there will be no negative impact of their work.

**Opportunities For Improvement:**

1. In the main manuscript, the author takes too much space to discuss why they need to have such experimental settings and how they set the experiments. The discussions of the experimental results are weak. Maybe this paper is more suitable to submit to a journal.
2. "In semi-supervised learning, which is to train a fixed encoder-model, pre-encode the datasets  and then only train a single linear layer as classifier." It can be computationally efficient, but it will not fully exploit the strengths of deep learning in an active learning scenario. A more typical and potentially more effective approach in deep active learning involves training the feature extractor alongside the classifier to ensure that the representations are continuously adapted to the newly labeled data.

**Relation To Prior Work:**

They discussed how this work differs from previous contributions but needs to be refined. Can see the "Review" section.

**Summary And Contributions:**

This paper introduced an active learning benchmark covering multiple domains (cv, nlp, and tabular dataset) with enough repetitions (50 runs per experiment). They demonstrate the importance of conducting multiple experimental runs to obtain meaningful conclusions.

---

> ### Author Rebuttal · Authors · 2024-08-14
>
> We thank the reviewer for their detailed review and would like to address the individual points:
> 1. Although the author states that the proposed benchmark is the first one that contains multiple domains, however, in each domain, many related empirical studies focus on specific domain, e.g., [r1], [r2], [r3], which weakens the contribution of this paper.
> - We will include [r1] into Table 1 and our related work ([r2] and [r3] do not provide experiments on their own).
> AL benchmarks are still riddled with comparability issues, because their choice of datasets, domains and models often is completely disjoint ([r2] Sec. 4.1 and [r3] Tab. 1).
> In our opinion, we need a benchmark that spans all domains and the most commonly used datasets and models to fix this problem.
> As described in our Section 5, we selected our datasets for maximum overlap with previous works. The same is true for our model selection, which either uses common models like ResNet18 or trivial to reproduce models, like MLPs.
> 2. The insights (e.g., the importance of multiple runs) build on existing knowledge in the active learning field.
> - Even though it is common knowledge that repeating an experiment enough times is crucial for ML research, in the field of Active Learning, we still observe many authors not doing so.
> To the best of our knowledge, we are the first to provide insight into exactly how often an AL experiment should be repeated in order to generate reliable results.
> The same is true for experimenting across different domains. Everyone knows that algorithms should be tested in as many domains as possible, but few actually apply that knowledge.
> 3. This paper takes too much space on the greedy oracle algorithm and seeding strategies, which may be complex for some readers to fully get the meaning.
> - We strongly believe that benchmark papers with a strong focus on technical detail are needed in the current state of AL literature.
> We discuss this point in detail in our answer for 4.
> 4. In the main manuscript, the author takes too much space to discuss why they need to have such experimental settings and how they set the experiments. The discussions of the experimental results are weak. Maybe this paper is more suitable to submit to a journal.
> - As you have mentioned in your review, there are a fair number of benchmarks for AL already.
> Each of those focus on 1-2 domains and place high emphasis on their results.
> However, apart from [1], each benchmark has the experimental setup and technical details on low priority, leading to a situation where everyone is doing their own framework in isolation and an informed discussion about best practices is very difficult.
> We hope (in combination with [1]) to open the door for such discussion in the area of AL.
> Nonetheless, we will extend discussion of our results.
> Based on the comment from another reviewer, we plan to study our results from the dimension of class imbalance, which currently is missing in our work.
> 5. Implementation of semi-supervised learning in our benchmark
> - As described in Section 5.2, we are not interested in beating the SOTA performance on any dataset, but rather in providing a fast and low-variance framework.
> The same idea applies to our choice of semi-supervised learning.
> For that reason, we opted for pre-encoded datasets instead of more sophisticated approaches.
> Finally, the authors of [2] provide good evidence that the ranking of algorithms does not change when the semi-supervised training changes (at least for the image domain).
>
> References: \
> [1]: Ji, Yilin, et al. "Randomness is the root of all evil: more reliable evaluation of deep active learning." Proceedings of the IEEE/CVF Winter Conference on Applications of Computer Vision. 2023. \
> [2]: Zhang, Jifan, et al. "LabelBench: A Comprehensive Framework for Benchmarking Adaptive Label-Efficient Learning." Journal of Data-centric Machine Learning Research (2024). \
> [r1]: Bahri, Dara, et al. "Is margin all you need? An extensive empirical study of active learning on tabular data." arXiv preprint arXiv:2210.03822 (2022). \
> [r2]: Zhang, Zhisong, Emma Strubell, and Eduard Hovy. "A survey of active learning for natural language processing." arXiv preprint arXiv:2210.10109 (2022). \
> [r3] Schröder, Christopher, and Andreas Niekler. "A survey of active learning for text classification using deep neural networks." arXiv preprint arXiv:2008.07267 (2020).

---

> > ### Comment · Reviewer_x7yL · 2024-08-26
> > **response**
> >
> > Thanks for the authors' effort, I would raise my score to 6.

---

> ### Author Response · Authors · 2024-08-26
> **Thank you for your feedback**
>
> We again thank the reviewer for the valuable feedback and hope that we have addressed all concerns. As the author-discussion deadline approaches, please let us know if further clarification is needed.
> Best regards,
> The authors"

---

### Official Review · Reviewer_3W2R · 2024-07-26
**Important and thorough contribution to active learning benchmarking**

**Rating:** 7
**Confidence:** 3
**Correctness:** As far as I can tell, yes
**Clarity:** see above

**Review:**

Overall this is a solid submission and provides a significant contribution to the active learning community. It simultaneously presents a practically useful framework while highlighting interesting and important empirical trends in active learning benchmarking.

Quality: overall this is generally a high quality submission. The appendix is thorough, there is an included codebase (that I have not reviewed in detail), and there are clever insights into active learning evaluation. The tables and figures in the paper are generally very helpful in illustrating the authors' key arguments including presenting a more comprehensive AL benchmark than previous work (table 1), high potential variability in active learning methods (figure 1), active learning dataset/model hyperparameters (table 2), comprehensive rank testing between methods (table 3 and figure 2), and two very clever synthetic datasets (figure 3) designed to stress test uncertainty and diversity based AL respectively.

Clarity: overall the manuscript was clear, although there are several places where the writing could be improved and I'd like to see these changes addressed (see below).

Originality/Significance: the desperate need for more comprehensive AL benchmarking is well understood in the community, and there are opportunities for better benchmarking. This paper makes strides in this direction, most significantly by addressing the question of multi-modal testing in a single framework. It does a good job of citing prior work. However this work should probably cite and discuss

Zhang, Jifan, et al. "LabelBench: A Comprehensive Framework for Benchmarking Adaptive Label-Efficient Learning." Journal of Data-centric Machine Learning Research (2024).

...which addresses some of the points discussed in this current submission (e.g., semi-supervised learning). The authors should consider adding LabelBench to the related work and Table 1.

**Strengths:**

see above

**Additional Feedback:**

n/a

**Documentation:**

yes

**Limitations:**

yes

**Opportunities For Improvement:**

Some aspects of the writing/presentation of this paper can be improved:
- The "oracle" needs to be explained earlier. It is discussed and referenced a lot, but you don't actually describe what it is until section 6. I was very confused, since typically "oracle" might also refer to an annotator. Things made sense at section 6, but a preview of what exactly the oracle does should be discussed as soon as possible. Looking again, it is discussed at line 53 but I actually found this paragraph very vague.
- what is "sngl" in Table 1? Single point sampling per round?
- what is 9(14) in Table 1 in the last row?
- not sure what $i \in [1 ... s]$ means really after line 93
- I think Figure 1 needs better descriptions and a better legend. I think I get the main point but it's confusing. E.g., "True mean" is vague, what is that the true mean of? Purple curve doesn't even appear on the legend.
- the choice of validation on the entire dataset needs more discussion (line 194). This is a huge criticism of active learning research, to choose parameters based on a full validation set. I understand the argument for lower variance in research evaluation, but I don't think the justification here is sufficient. In particular, Figure 1 argues that the high variance in research results is a problem. Why mask it, with an unrealistic validation? Won't that make things worse?


Minor:
- spurious number "1" in line 33
- wether -> whether line 36
- extra space line 171

**Relation To Prior Work:**

see above

**Summary And Contributions:**

This work presents a new benchmark and codebase for evaluating active learning algorithms systematically across multiple data domains. Most active learning algorithm evaluation has focused solely on the image domain, and other works that consider non-image domains do not comprehensively evaluate their methods across data modalities. Instead, this work presents a single framework for evaluating various active learning strategies across several domains. The contributions include showing significant variation in low-repetition active learning evaluation, demonstrating the need for more evaluation runs; a new baseline oracle method to upper bound methods' performance; the CDALBench benchmark presenting a consistent pipeline to evaluate AL methods across domains along with two new synthetic evaluation datasets.

---

> ### Author Rebuttal · Authors · 2024-08-14
>
> We thank the reviewer for their detailed review and would like to address the individual points:
> 1. This work should probably cite and discuss Zhang et al.
> - This is a great hint. We will add Zhang et al. to Table 1 and our Related Work
> 2. The "oracle" needs to be explained earlier. It is discussed and referenced a lot, but you don't actually describe what it is until section 6. [...]
> - We will insert the following sentence in line 55 to clarify the main idea of our oracle algorithm early on:
> "Our oracle relies on directly testing a small sample of points in every iteration if they induce an improvement in test accuracy and selects the optimal point from that small sample."
> 3. what is "sngl" in Table 1? Single point sampling per round?
> - Yes, we will improve the clarity in Table 1 by not using an abbreviation for "single"
> 4. what is 9(14) in Table 1 in the last row?
> - We have 9 datasets in our benchmark, 5 of which have a pre-encoded version (excluding text and synthetic) , which brings the total number of experiments to 14.
> We will include this description in the caption of Table 1.
> 5. not sure what i\in means really after line 93
> - It simply indexes all points in the unlabeled set.
> We will improve the clarity here as well
> 6. I think Figure 1 needs better descriptions and a better legend. [...]
> - We will add all visible curves to the legend in Figure 1.
> Additionally, we will revise the caption of Figure 1 to make it more accessible for the readers.
> 7. the choice of validation on the entire dataset needs more discussion (line 194) [...]
> - It is true that, by proposing to use a fully labeled validation set, we implicitly claim that most AL research so far had been flawed.
> However, we don't want to challenge AL literature as a whole, but rather push for a stronger separation of AL research (with validation set) and AL applications (without validation set).
> With our approach, we are not trying to mask the high variance; instead, we argue that **due** to the high variance, we should fully optimize our hyperparameters on validation to avoid exacerbating its effects.
> The core hypothesis is that a top-performing algorithm in an experiment with good hyperparameters also performs well in the application case with bad/worse hyperparameters.

---

> > ### Comment · Reviewer_3W2R · 2024-08-14
> >
> > Thank you for your comments. I am satisfied with these answers and maintain my score as an accept. I think it would strengthen the paper to include discussion along the last point made in particular, discussing the role of the validation set. This hypothesis about the strongest fully-optimized active learning method in research translating well to applications should also be discussed. I don't think the authors need to solve bridging that gap in this paper, but it's actually not clear to me if that hypothesis is true. It may be the case that in applications, the best performing active learning method on fully optimized validation data in research, may be extremely brittle and sensitive to hyperparameters in practice. In practice, it may be better to adopt a method that does not perform the best in a research setting, but is more robust to the selection of hyperparameter choice in practice. This is what most practitioners would agree with I'm sure. Can the authors comment about this sensitivity to hyperparameters as it relates to their core hypothesis?

---

> > > ### Author Response · Authors · 2024-08-15
> > >
> > > This entire topic is well worth its own survey paper.
> > > In our opinion, current strands of AL methods are mostly concerned with reducing uncertainty in the model.
> > > Given bad hyperparameters, a model either learns extremely slow or is prone to collapsing.
> > > In these cases, uncertainty-based AL methods would simply sample more points from the same regions and slowly recover model performance.
> > > The same is true for clustering approaches like CoreSet, since all of them use the current model to embed the points first.
> > > In such a setup, where the model is prone to collapsing, all AL methods should suffer the same.
> > > The only type of AL method that might be disproportionately disadvantaged would be methods that rely on expected model change (EMC) or directly optimize test accuracy (like meta-learned RL agents).
> > > Even though EMC is not currently the main focus in AL and RL approaches still remain hard to reproduce, a systematic study on the topic of hyperparameter sensitivity is necessary in the near future to preempt problems of this nature.

---

### Official Review · Reviewer_iCXF · 2024-07-28
**This paper presents a benchmark for evaluation of active learning methods. This is the first benchmark to include tasks over three different domains (computer vision, NLP, tabular) and is designed in such a way that a large number of runs can be done for each experiment.**

**Rating:** 7
**Confidence:** 3
**Correctness:** Yes
**Clarity:** Yes

**Review:**

The benchmark proposed in this paper is the first to include tasks over three different domains (computer vision, NLP, tabular) and is designed in such a way that a large number of runs can be done for each experiment. The authors use this benchmark to thoroughly evaluate many methods existing in the literature; and the evaluation reveals that methods that were supposed to be state-of-the-art based on evaluations on other datasets have poor performance in this benchmark and are sometimes even worse than random. These findings have the potential of sparking discussions on what it means for an active method to be consistently good across multiple domains, which in turn can lead to better methods.

On the other hand, other contributions of the paper are relatively weaker. First, the results on the synthetic benchmarks are unsurprising and the contribution is unclear since they show that methods that focus on uncertainty sampling like margin fail on distributions that are adversarially designed to make them fail. Second, the algorithm for the greedy oracle has no clear guarantees, the design choices are unclear and the description is hard to follow. Finally, the evaluation is done with relatively small models and datasets and it is unclear whether it translates to larger ones.

**Strengths:**

- The benchmark is more comprehensive than previous ones existing in the literature. In particular, it is the first to include tasks over three different domains (computer vision, NLP, tabular) and is designed in such a way that a large number of runs can be done for each experiment.

-  The authors use this benchmark to thoroughly evaluate many methods existing in the literature; and the evaluation reveals that methods that were supposed to be state-of-the-art based on evaluations on other datasets have poor performance in this benchmark and are sometimes even worse than random.

- The finding in this paper have the potential of sparking discussions on what it means for an active method to be consistently good across  multiple domains, which in turn can lead to better methods.

**Additional Feedback:**

-

**Documentation:**

Yes

**Limitations:**

Yes

**Opportunities For Improvement:**

- The results on the synthetic benchmarks are unsurprising and the contribution is unclear since they show that methods that focus on uncertainty sampling like margin fail on distributions that are adversarially designed to make them fail.

- The algorithm for the greedy oracle has no clear guarantees, the design choices are unclear and the description is hard to follow.

- The evaluation is done with relatively small models and datasets and it is unclear whether it translates to larger ones.

**Relation To Prior Work:**

Yes

**Summary And Contributions:**

This paper presents a benchmark for evaluation of active learning methods. This is the first benchmark to include tasks over three different domains (computer vision, NLP, tabular) and is designed in such a way that a large number of runs can be done for each experiment. The authors use this benchmark to thoroughly evaluate many methods existing in the literature. They also present synthetic datasets and an algorithm for oracle evaluation that is more efficient than those proposed in previous work.

---

> ### Author Rebuttal · Authors · 2024-08-14
>
> We thank the reviewer for their detailed review and would like to address the individual points:
> 1. The results on the synthetic benchmarks are unsurprising and the contribution is unclear [...]
> - Our primary goal is to validate these approaches through a targeted ablation study, highlighting the strengths and limitations of both types of algorithms.
> You are correct when you say that algorithms like margin sampling behave exactly as expected, but in our opinion, the behavior of a hybrid algorithm like BADGE was completely untested in these adversarial environments (in the case of BADGE, we revealed weaknesses akin to vanilla uncertainty sampling methods, even though BADGE uses an clustering-like algorithm to select its samples).
> These datasets mainly serve as a resource for benchmarking future AL methods, particularly for hybrid approaches that use ideas from uncertainty sampling and clustering/diversity methods.
> 2. The algorithm for the greedy oracle has no clear guarantees, the design choices are unclear and the description is hard to follow
> - We designed our oracle based in the following two ideas:
>     - Have an algorithm that does not rely on search or excessive enumeration, so it can be computed efficiently
>     - Pose a performance that is "good enough", so that it beats every AL algorithm on average without wasting computational resources for diminishing lifts
> - Since we are mainly interested in beating other AL algorithms and not a theoretically optimal upper bound we did not see the need derive a performance guarantee for our oracle.
> - To improve the description of the oracle, we will add comments to the pseudocode in Appendix H and reference it more clearly in the main body:
> Remove Line 258: The pseudocode for our oracle can be found in App. H.
> Add: For a detailed description of the oracle algorithm and how it is used in the AL loop, please refer to App. H.
>
> 3. The evaluation is done with relatively small models and datasets and it is unclear whether it translates to larger ones.
> - We agree that no comparable benchmark has shown a systematic conversion of small-model setup to large-model setups.
> However, we have evidence in domain-specific literature that supports our hypothesis that our ranking transfers to larger models:
>     - [4] use larger image models like ViT-B32 and also find least confidence sampling and BADGE to be the best algorithms for images
>     - [5] use BERT models in the text domain and also find BADGE to be the best (they don't evaluate margin sampling)
>     - [6] use considerably larger MLPs for the tabular domain than we do, but also find margin sampling to be consistenly top-performing
>
> References: \
> [4]: Zhang, Jifan, et al. "LabelBench: A Comprehensive Framework for Benchmarking Adaptive Label-Efficient Learning." Journal of Data-centric Machine Learning Research (2024).\
> [5]: Rauch, Lukas, et al. "Activeglae: A benchmark for deep active learning with transformers." Joint European Conference on Machine Learning and Knowledge Discovery in Databases. Cham: Springer Nature Switzerland, 2023.\
> [6]: Bahri, Dara, et al. "Is margin all you need? An extensive empirical study of active learning on tabular data." arXiv preprint arXiv:2210.03822 (2022).

---

> ### Author Response · Authors · 2024-08-26
> **Thank you for your feedback**
>
> We again thank the reviewer for the valuable feedback and hope that we have addressed all concerns. As the author-discussion deadline approaches, please let us know if further clarification is needed.
> Best regards,
> The authors"

---

### Decision · Program_Chairs · 2024-09-26

**Decision:**

Accept (Poster)

**Comment:**

CDALBench is the first active learning benchmark comprising tasks in CV, NLP, and tabular data. They also include an evaluation of existing methods, showing that different active learning strategies perform best in different settings, an unsurprising but nonetheless valuable result. While reviewers were less optimistic about the quality of the model evaluations included, the reviewers were unanimous in their evaluation of the benchmark itself, which should be a good resource to model developers, and their scores reflect that with a consistent recommendation for acceptance. I echo this opinion - it would have been nicer to see a stronger empirical study alongside the benchmark, but as it stands the benchmark will be of good utility to the broader community.